



# Snow Ensemble Uncertainty Project (SEUP): Quantification of snow water equivalent uncertainty across North America via ensemble land surface modeling

Rhae Sung Kim[1,2], Sujay Kumar[1], Carrie Vuyovich[1], Paul Houser[3], Jessica Lundquist[4], Lawrence Mudryk[5], Michael Durand[6], Ana Barros[7], Edward J. Kim[1], Barton A. Forman[8], Ethan D. Gutmann[9], Melissa L. Wrzesien[1,2], Camille Garnaud[10], Melody Sandells[11], Hans-Peter Marshall[12], Nicoleta Cristea[4], Justin M. Pflug[4], Jeremy Johnston[3], Yueqian Cao[7], David Mocko[1,13], Shugong Wang[1,13]

[1]Hydrological Sciences Laboratory, NASA Goddard Space Flight Center, Greenbelt, MD, USA
[2]Universities Space Research Association, Columbia, MD, USA
[3]Department of Geography and Geoinformation Sciences, George Mason University, Fairfax, VA, USA
[4]Civil and Environmental Engineering, University of Washington, Seattle, WA, USA
[5]Climate Research Division, Environment and Climate Change Canada, Toronto, Ontario, Canada
[6]School of Earth Sciences and Byrd Polar and Climate Research Center, The Ohio State University, Columbus, OH, USA
[7]Civil and Environmental Engineering, Duke, University, Durham, NC, USA
[8]Civil and Environmental Engineering, University of Maryland, College Park, MD, USA
[9]National Center for Atmospheric Research, Boulder, Colorado, USA
[10]Meteorological Research Division, Environment and Climate Change Canada, Dorval, Quebec, Canada
[11]Geography and Environmental Sciences, Northumbria University, Newcastle upon Tyne, UK
[12]Department of Geosciences, Boise State University, Boise, ID, USA
[13]Science Applications International Corporation, Reston, VA, USA

*Correspondence to*: Rhae Sung Kim (rhaesung.kim@nasa.gov)

**Abstract.** The Snow Ensemble Uncertainty Project (SEUP) is an effort to establish a baseline characterization of snow water equivalent (SWE) uncertainty across North America with the goal of informing global snow observational needs. An ensemble-based modeling approach, encompassing a suite of current operational models, is used to assess the uncertainty in SWE and total snow storage (SWS) estimation over North America during the 2009-2017 period. The highest modeled SWE uncertainty is observed in mountainous regions, likely due to the relatively deep snow, forcing uncertainties, and variability between the different models in resolving the snow processes over complex terrain. This highlights a need for high-resolution observations in mountains to capture the high spatial SWE variability. The greatest SWS is found in Tundra regions where, even though the spatiotemporal variability in modeled SWE is low, there is considerable uncertainty in the SWS estimates due to the large areal extent over which those estimates are spread. This highlights the need for high accuracy in snow estimations across the Tundra. In mid-latitude boreal forests, large uncertainties in both SWE and SWS indicate that vegetation-snow impacts are a critical area where focused improvements to modeled snow estimation efforts need to be made. Finally, the SEUP results indicate that SWE uncertainty is driving runoff uncertainty and measurements may be beneficial in reducing uncertainty in SWE and runoff, during the melt season at high latitudes (e.g., Tundra and Taiga regions) and in the Western mountain regions, whereas observations at (or near) peak SWE accumulation are more helpful over the mid-latitudes.



## 1 Introduction

Seasonal snow plays an important role in Earth's climate and ecological systems and influences the amount of water resources available for agriculture, hydropower, and human consumption, serving as the primary freshwater supply for more than a billion people worldwide (Foster et al., 2011). There is a critical need to better understand the role of snow in global climate

and land-atmosphere interactions (Brooks et al., 2011; Robinson and Kukla, 1985; Stielstra et al., 2015), and associated influences on soil moisture, vegetation health, and streamflow (Berghuijs et al., 2014; Ryberg et al., 2016; Stewart et al., 2005). Decreases in total snow water storage can lead to increased droughts (Barnett et al., 2005; Fyfe et al., 2017; Mahanama et al., 2012) and wildfires (Westerling et al., 2006). In addition, snowmelt is a dominant driver of flooding in many regions of the U.S. (Berghuijs et al., 2016).

Though accurate and timely estimates of snow water equivalent (SWE) are required for water and ecosystem management, obtaining reliable, spatially distributed SWE has been a challenge, particularly at continental and global scales. At these scales, satellite observations are ideal, but global SWE observations remain a major gap in snow remote sensing (Dietz et al., 2012; Lettenmaier et al., 2015; Nolin, 2010), and the U.S. National Research Council committees of the Decadal Survey (National Academies of Scienecs, Engineering, and Medicine, 2018) identifies SWE as a missing component of spaceborne water cycle

measurements. This has motivated the advancement of models and remote sensing techniques to estimate global snow characteristics (e.g., NASA SnowEx, Durand et al. 2019; Kim et al., 2017). Developing the necessary observational methods for global coverage while also supporting local snow applications is a significant challenge facing the snow community (Dozier et al., 2016; Lettenmaier et al., 2015). Both models and remote sensing techniques are impacted by numerous factors, resulting in significant spatial or temporal errors in SWE estimation.

A potential solution to reduce uncertainty associated with any single technique is to combine models and remote sensing in a data assimilation framework, but this requires an understanding of the underlying uncertainty to be employed. In this study, called the Snow Ensemble Uncertainty Project (SEUP), we apply an ensemble-based land surface modeling approach to establish a baseline characterization of SWE and its corresponding uncertainty across North America. The term "SWE uncertainty" used in this study will refer to the range of SWE estimates across models and is quantified as the ensemble spread.

Compared to the use of a single model realization, ensemble modeling is generally considered a better approach to characterize the inherent uncertainties in modeling, with the ensemble spread providing a measure of the uncertainty in the predictions across models and forcing data (Bohn et al., 2010; Dirmeyer et al., 2006; Franz et al., 2008; Guo et al., 2007; Kumar et al., 2017; Mitchell et al., 2004; Mudryk et al., 2015; Murphy et al., 2004; Xia et al., 2012). An ensemble evaluation can also lead to increased skill by combining a variety of model estimates and allowing the individual model errors to cancel each other out

(Xia et al., 2012). We use the ensemble SWE estimates to assess the general spatial and temporal North American SWE characteristics.

The SEUP ensemble is comprised of 12 ensemble members, created by the combination of four different land surface models (LSMs) and three different forcing datasets. By using a mix of different LSMs and boundary conditions, the SEUP





ensemble captures the uncertainties from both these sources. The design of the SEUP ensemble is focused on current snow capabilities in macroscale modeling, as the land models and forcing datasets selected here represent models and datasets currently being employed at key operational centers and systems (described in Section 2.2). The designed experiment is conducted at a 5km spatial resolution for multiple winter snow seasons (2009-2017). By using a range of forcing products and commonly-used operational models, we assume that the SEUP ensemble implicitly provides a representation of both sources of uncertainty. It is likely, however, that the SEUP ensemble may be deficient in representing the "true uncertainty," given the possible errors in boundary conditions, model parameters, and model structure. Nevertheless, the SEUP ensemble establishes an important baseline over the continental scales to characterize current capabilities and inform global snow observational requirements. Toward this goal, in this article we strive to address several gaps in our current understanding of SWE uncertainty with our simulation of snow states over the North American continental domain, including: 1) *Where are the areas of significant uncertainty in SWE?*, 2) *What is the seasonality of SWE uncertainty and its spatial distribution?*, 3) *How does uncertainty in SWE vary with key land surface characteristics such as vegetation, topography, and snow climate?*, 4) *How do these regions of high SWE uncertainty correlate with runoff uncertainties?*.

The paper is organized as follows: Section 2.1 introduces the study area and time period, followed by the descriptions of the LSMs and forcing datasets used in this study in Sections 2.2 and 2.3, respectively. Section 2.4 provides the details about the experimental design, ensemble-based methods, and datasets used in the uncertainty evaluation. An evaluation of the SEUP ensemble against a number of reference products is presented in Section 3.1. Section 3.2 provides the results of SWE uncertainty analysis over North America. The influence of factors such as topography, snow regime, and vegetation type on SWE/SWS uncertainty is examined in Section 3.3. Section 3.4 discusses how the snow modeling uncertainty impacts the uncertainty in the terrestrial water budget components. Finally, Section 4 provides the major findings and conclusions of this effort.

## 2 Study area and ensemble configuration

### 2.1 Study area and time period

The study area is the North American continental domain consisting of a 0.05° latitude by 0.05° longitude equidistant cylindrical grid that extends from 24.875°N to 71.875°N and 168.625°W to 51.875°W (Fig. 1a). The glacier regions are excluded from the study domain as the representations of glacier processes are limited in the LSMs used here. These glacier exclusions are developed using the Global Land Ice Measurement from Space (GLIMS) geospatial glacier database (Raup et al., 2007). The model integrations and analyses are performed from 2000 to 2017 with the first 9 years (2000-2009) used as a model spin-up to initialize the model's thermal and hydraulic equilibrium states.

### 2.2 Land Surface Models (LSMs)



The National Aeronautics and Space Administration (NASA) Land Information System (LIS; Kumar et al., 2006; Peters-
        Lidard et al., 2007) is a comprehensive terrestrial modeling infrastructure designed to facilitate the efficient use and
        assimilation of terrestrial observations. This study uses a modeling configuration within the NASA LIS that employs four
        different land surface models (LSMs) of varying complexity at a 5km spatial resolution over North America: 1) Noah version
        2.7.1 (Noah2.7.1, Ek et al., 2003), 2) Noah-MP version 3.6 (Noah-MP3.6, Niu et al., 2011; Yang et al., 2011), 3) Catchment
version 2.5 (CLSM-F2.5, Ducharne et al., 2000; Koster et al., 2000), and 4) Joint UK Land Environment Simulator (JULES,
        (Best et al., 2011; Blyth et al., 2006; Clark et al., 2011). These models are selected because all are used operationally at major
        modeling centers (e.g., Noah2.7.1 is used at the U.S. National Centers for Environmental Prediction (NCEP), Noah-MP3.6 at
        the National Water Model (NWM), CLSM-F2.5 at the NASA Global Modeling and Assimilation Office Research (GMAO),
        and JULES at the United Kingdom Met Office (UKMO)) to provide a baseline of current operational capabilities. Though the
outputs from these models are used widely for a variety of water resources management applications, only a few studies have
        conducted careful examination of their differences and limitations, particularly over continental spatial scales.

        All four LSMs are able to dynamically predict land surface water and energy fluxes in response to surface meteorological
        forcing inputs, but they differ in their structural representation of surface and subsurface water, and energy balance processes.
        As most land surface models were originally developed to provide the lower boundary conditions for global atmospheric
models, their applicability is largely assumed to be at coarse spatial scales where the influence of lateral interactions is
        negligible. Consequently, similar to other model physics components, the snow physics schemes in these models are not
        designed to resolve processes at fine spatial scales (e.g., < 100 m), such as the influence of blowing and drifting snow. Further,
        the complexity of snow metamorphic process representation varies across these models. The snow schemes in these models
        range from a simple single-layer scheme in both Noah2.7.1 and JULES to three-layer intermediate complexity schemes in both
Noah-MP3.6 and CLSM-F2.5, which greatly influence the snowpack thermodynamics and the resulting timing and presence
        of melt (Dutra et al., 2011). It is important to note that initial model conditions and model parameters used in the operational
        set-up were not tuned in this study to assess current configurations (Best et al., 2011, Blyth et al., 2006, Clark et al., 2011,
        Ducharne et al., 2000, Ek et al., 2003, Koster et al., 2000, Niu et al., 2011, Yang et al., 2011). The key details of the model
        configurations with forcing datasets are summarized in Table S1.

**2.3 Forcing datasets**

        Three different modern forcing datasets are used to drive the models: 1) Modern-Era Retrospective Analysis for Research and
        Applications, version 2 (MERRA2; Gelaro et al., 2017; Molod et al., 2015), 2) Global Data Assimilation System (GDAS;
        Derber et al., 1991), and 3) European Centre for Medium-Range Weather Forecasts (ECMWF; Molteni et al., 1996). All
        models are run at 15-minute time intervals. To improve the spatial representativeness of the coarse resolution meteorological
inputs, the input forcing fields were downscaled to a 5 km grid as follows. Meteorological inputs of near surface air
        temperature, relative humidity, surface pressure, wind, and downward longwave radiation are downscaled by applying a lapse-
        rate and hypsometric adjustments using the 5 km Shuttle Radar Topography Mission (SRTM) elevation data. The lapse-rate



correction method follows the approach used in the North American Land Data Assimilation System (NLDAS)-1 and 2 projects (Cosgrove et al., 2003), where a static environmental lapse rate of 6.5 K/km is used to apply an elevation adjustment to the

coarse meteorological fields. The downwelling shortwave radiation fields are downscaled using terrain characteristics of slope and aspect as described in Kumar et al. (2013). Over the east and west-facing slopes, the slope and aspect-based corrections lead to improvements to diurnal processes. Kumar et al. (2013) demonstrated that these adjustments are particularly important for improving snow simulations over midlatitude domains in complex topography. The precipitation fields are downscaled using a variant of the scaling approach of Lenderink et al. (2007) with the high resolution monthly precipitation climatology

dataset, WorldClim (Fick and Hijmans, 2017). The downscaling is performed by fixing the ratio of high-resolution precipitation climatology to that of the same climatology at the coarser-scale resolution in order to maintain the heterogeneity of the precipitation forcing fields. The three global data sets are all derived using global atmospheric models that assimilate a large collection of surface and atmospheric observations and differ primarily in the atmospheric model and assimilation system used.

**2.4 Methods**

**2.4.1 SEUP ensemble evaluation methods**

We use two metrics to evaluate the SEUP ensemble: 1) ensemble rank (ER), which ranks the observation relative to the ensemble providing a measure of how well the ensemble encompasses a reference observation, and 2) continuous rank probability score (CRPS; Matheson and Winkler, 1976) that measures the difference between the model and the reference

distributions. For computing ER, the ensemble is first organized in the following order: CLSMF-2.5 (ensemble members 1 to 3), JULES (4 to 6), Noah-MP3.6 (7 to 9), and Noah2.7.1 (10 to 12), with the order within each LSM being the runs forced with ECMWF, GDAS, and MERRA2 data, respectively. The ensemble SWE at each grid point and each temporal instance is then sorted and ordered first. The rank of the reference data within this sorted array is then used as the ER. If the observation is more than 10% of the highest ensemble member, then the rank is set to 13. As a demonstrative example, if the ensemble

SWE values are 1, 3, 7, 2, 4, 5, 6, 1, 3, 8, 1, 0 units and the observation has a value of 5 units, the ER of the observation is set to 9 as the sorted array will be 0, 1, 1, 1, 2, 3, 3, 4, 5, 6, 7, 8. Note that the main objective of the ER metric is to examine whether the ensemble encompasses the reference data.

CRPS is an often-used performance measure in probabilistic forecasting, computed using Eq. (1). It provides a measure of the degree of difference between the model distribution and the observation. CRPS reduces to the mean absolute error when

used with deterministic (single-member) ensembles.

$$CRPS = \int_{X=-\infty}^{X=+\infty} (P_m - P_o)^2 dx \ , \tag{1}$$

where $P_m$ represents the cumulative distribution function (CDF) of the model and $P_o$ represents the observation occurrence. Note that the SEUP ensemble size (12) is relatively small, which may affect the resolution of the CDF derived from it. Nevertheless, CRPS provides an integrated way of capturing the error associated with the SEUP ensemble when compared to



reference measurements, where a low (good) score indicates small ensemble spread that encompasses the reference observation and a high (bad) score indicates large spread and/or large difference from the observation.

### 2.4.2 Reference and ancillary datasets used in the uncertainty evaluation

The reference datasets used for evaluation in SEUP are: (1) the daily, gridded snow depth, and SWE analysis from the NOAA National Weather Service's National Operational Hydrologic Remote Sensing Center (NOHRSC) SNOw Data Assimilation

System (SNODAS; Barrett, 2003) available at 30 arcseconds spatial resolution, (2) daily gridded estimates of snow depth and SWE developed by University of Arizona (UA; Zeng et al., 2018), and (3) the daily, gridded snow depth analysis from the Canadian Meteorological Centre (CMC; Brown and Brasnett, 2010) available at 25 km spatial resolution. All three datasets are model-based, but they incorporate in-situ measurements from various ground networks. SNODAS analyses also encompass satellite and airborne measurements, meteorological aviation reports, and special aviation reports from the World

Meteorological Organization (WMO). Though these data are subject to errors, this product provides a consistent, spatially distributed estimate of snowpack conditions throughout the U.S. and has been used as a comparison dataset in numerous studies (Guan et al., 2013; Meromy et al., 2013; Vuyovich et al., 2014). The UA analysis is developed using an empirical temperature index snow model with data from networks such as the National Resources Conservation Services' SNOTEL and the National Weather Service's Cooperative Observer Program (COOP). The dataset was developed to provide a high-resolution, long-term

snow mass product for use in assessing climate change impacts (Zeng et al. 2018). SNODAS and UA datasets are available only over the continental U.S., whereas the CMC data are used for snow evaluation over the entire domain. While the CMC data have been frequently used for LSM evaluation (Forman et al., 2012; Reichle et al., 2017; Takala et al., 2011), and have been shown to capture interannual variability well (Brown et al., 2018), several studies have provided evidence that the data underestimate SWE (Dawson et al., 2016; Wrzesien et al., 2017). Since CMC data only includes snow depth, we evaluate the

modeled snow depth fields (instead of SWE, when comparing with CMC data) for the sake of uniformity.

A number of ancillary datasets representing topography, vegetation type, and snow class are used in stratifying the spatial dependence of snow uncertainty. First, to treat mountainous and non-mountainous regions separately in our study, we upscale Wrzesien et al. (2018)'s 1km binary mountain mask to our 5km grid (see Fig. 1b). Wrzesien et al. (2018) adopted the definition of "mountain" from Kapos et al. (2000) based on the elevation, slope, and local relief. In their work, the mask was divided into

eleven individual mountain domains, which we use here to evaluate SEUP results over mountain areas. Table S2 shows the areas of these eleven individual mountain ranges.

An uncertainty analysis on SWE estimation is performed across different snow class regions to understand which regions account for the highest variability. To the best of our knowledge, analyzing uncertainty in SWE estimation across different snow classes at continental scales has not been explored in the literature. In this analysis we use a snow classification proposed

by Sturm et al. (1995), and recently updated at a higher (10 km) resolution (Liston and Sturm, 2014), which analyzes the relationships among textural and stratigraphic characteristics of snow layers, climate variables (e.g., air temperature, precipitation, and wind speed), and vegetation to globally categorize terrestrial snow into seven classes: Tundra, Taiga,





Maritime, Ephemeral, Prairie, Warm forest, and Ice. We downscale this global snow classification dataset to our 5km model grid (from the native 10km spatial resolution). Figure 1c shows the individual domains of 7 snow classes over North America

and Table S3 presents their individual areas.

The Moderate Resolution Imaging Spectroradiometer (MODIS)-derived land cover employing the International Geosphere-Biosphere Programme (IGBP) land cover classification method is used to examine the influence of SWE uncertainty to vegetation. For simplicity of comparison, we reclassify the original 20 different land cover classes into 2 classes. These reclassified land cover classes (i.e., forested vs non-forested) are displayed in Fig. 1d, and their areas are presented in

Table S4.

## 3 Results and discussion

This section presents and discusses results from a range of perspectives. Section 3.1 compares the ensemble with observations derived from data assimilation techniques. Section 3.2 considers spatial and temporal variation in model uncertainty. Ensemble characteristics are linked to land surface classification in Section 3.3. Finally, the impact of model uncertainty on runoff

estimation is examined in Section 3.4.

### 3.1 Evaluation of the SEUP ensemble

To evaluate the snow estimates from the SEUP ensemble, three available reference products (described in Section 2.4.2) are used. Figure 2 shows maps of average Ensemble Rank (ER) and average Continuous Rank Probability Score (CRPS) (see Section 2.4.1) for the SEUP ensemble compared to three reference datasets during the time period of 2009 to 2017. The

examination of ER indicates that in general the SEUP ensemble encompasses the three reference measurements. In the SNODAS comparison, ER values larger than 12 can be seen in regions with larger snowpacks, such as the Rockies, indicating that over these areas the SEUP ensemble may be biased low. The ER patterns are similar in both SNODAS and UA comparisons, though the UA comparison shows more spatial variability across different latitudes. Over the high latitude regions in the CMC comparisons, the low end of the SEUP ensemble envelops the CMC data, whereas over the other parts of

Canada, the reference data is matched by the middle to higher end of the ensemble.

The CRPS comparison provides a measure of the discrepancies between the SEUP ensemble and the reference datasets. Over most of the domain, including the northeast/Midwest U.S. and high plains, the CRPS values are low, indicating a small ensemble spread that agrees with SNODAS and UA data. As expected, the largest CRPS values are observed over locations with deep snowpacks, such as the Rocky and Pacific coastal mountains, where the SEUP ensemble spread is greatest. Similar,

but more muted patterns of disagreement are seen with the CMC data compared to SNODAS and UA over mountainous regions, indicating that the SEUP simulations are more consistent with CMC in those areas. In the CMC comparison, larger errors are also observed at high latitudes, which are likely caused by a combination of larger uncertainties in the boundary conditions and model formulations. Relatively good agreement of SEUP with SNODAS and UA in the ER and CRPS-based assessments is particularly encouraging, as it provides a measure of confidence that the ensemble encompasses reality.





## 3.2 SWE uncertainty analysis

### 3.2.1 Spatial variability of SWE

An overall assessment of the SWE results is shown in Fig. 3, which presents the spatial distributions of ensemble mean SWE, the coefficient of variation of ensemble mean SWE, and the range of ensemble mean SWE. Because the seasonal timing of the greatest SWE, as we will show in Section 3.2.2 and the largest uncertainty in SWE differ substantially across the North American study domain, we first consider a simple annual mean averaged SWE across the entire time period. Seasonal timing of when the greatest uncertainty occurs is deferred to Section 3.2.2. For each pixel, the period-, annual-, ensemble mean SWE is computed by taking an average of 3-hourly SWE from 12 ensemble members over the entire study time period. We limit the range of coefficient of variation displayed from 0 to 1 (including no-snow time periods in the calculation) for reasons of visual clarity.

The largest spread in ensemble mean SWE is found in regions with the deepest snow (see Fig. 3a and 3c), particularly along the northern Pacific coastline. Eastern Canada along the northern Atlantic coastline and northern Rocky Mountains also show a high spread of SWE between ensemble members. These highly complex terrains have relatively high snowfall precipitation, and the large spread is likely due to different rain/snow partitioning schemes in each LSM. Similarly, the spatial distribution of the coefficient of variation shows larger values in areas with the higher ensemble mean SWE and ensemble spread. This indicates that the larger spread is not only due to the larger mean SWE in these areas. In addition, Fig. 3b also shows significant variability across the middle of North America, mostly collocated with boreal forest regions containing denser vegetation, indicating the handling of vegetation on SWE simulations as another source of dissimilarity among the SEUP ensemble members.

### 3.2.2 Seasonal variability of SWE

Figure 4 shows spatial maps of the peak SWE (panel a) and the highest SWE spread (panel b) along with characterizations of the seasonality of the SWE uncertainty (panels c and d). A measure of the spatial variability on the date of the highest SWE uncertainty is determined by computing the day of year (DOY) in each water year with the highest ensemble spread and then averaging DOY across the years to identify the times of high and low uncertainty in SWE over North America. This average DOY of highest spread is compared with the average DOY of the peak SWE to determine when the largest variability in the SWE spread occurs within the snow season. The DOY with the greatest SWE spread ranges from Dec-Apr time frame in the lower latitudes to May-June months in the high latitudes (Fig. 4c). In addition, the seasonality of the greatest SWE uncertainty at higher elevations, such as over the Rocky Mountains and the Pacific coastline, is shifted later in the season as compared to the lower elevation areas at the same latitude.

The largest SWE spread is along the northern Pacific coastline and eastern Canada along the northern Atlantic coastline (Fig. 4a). If the average DOY with the highest SWE spread matches that of the peak SWE, it suggests that the largest modeling uncertainty occurs in the peak winter time period. From Fig. 4d, we find that DOYs with the highest SWE spread and peak



SWE are very close to each other in the U.S.. Over Canada, the highest SWE spread has a later DOY than that from the peak SWE, indicating that the largest disagreements in the model estimates are during the melt season. One reason for this could be that the input meteorology has larger differences over high latitudes, whereas over the continental U.S., they are better

constrained due to the greater availability of ground and radar measurements, resulting in better agreement in the determination of snow melt regimes.

### 3.2.3 Interannual variability of SWE

We compare the time series of domain-averaged daily mean SWE for each ensemble to examine the temporal variability among the ensemble members (Fig. 5). Interestingly, the interannual variability in the peak SWE across the ensemble is small (see

Fig. 5), indicating that the simulated total snow water storage in North America as a whole did not change significantly year by year during this time period. Larger spread in the years of 2010 and 2011 are seen when comparing with other years. At a domain averaged scale, the largest spread in climatological SWE among the ensemble members is seen during the months of Feb to Apr and varies by as much as ~60%. In Fig. 5, variability due to model differences (e.g., between solid lines) is generally larger than variability due to forcing data (e.g., between blue lines), consistent with Broxton et al., (2016).

### 3.2.4 Impact between different LSMs and forcing data on SWE uncertainty

We further examine the influence of models and forcing data on SWE variability by comparing each ensemble grouped by LSMs and forcing data. Figure 6 shows the distribution of domain-averaged, annual mean SWE and indicates that there are smaller differences in SWE across the forcing datasets when driven with a common LSM, whereas larger differences are seen across the LSMs when driven with a common forcing data. This finding, from both temporal and spatial analyses, indicates

that within our ensemble set, the dominant factor driving uncertainty in SEUP SWE estimates over North America is from the LSM. This result is consistent with that from Mudryk et al. (2015) using an analogous but more limited ensemble of gridded snow products (cf. Fig. 12 in that paper). Note that both conclusions are based on analysis at the continental or hemispheric scale, and there could be differences at smaller scales and/or in topographically complex regions such as mountainous areas. For example, Raleigh et al. (2016) and Günther et al. (2019) showed the forcing data to be the primary driver of SWE

uncertainty in their study, which focused on a limited number of relatively small sites mostly in mountainous terrains. Similarly, Yoon et al. (2019) recently showed that the forcing data drove the uncertainty of model simulated estimates (i.e., precipitation, evaporation, and runoff) over High Mountain Asia, because of significant differences in the quality of reliable reference measurements over the domain. Future efforts should focus on evaluating model parameterizations and snow physics schemes such as sublimation, blowing and drifting snow, and snow-vegetation interactions to identify how representations of

snow physical processes are driving spread.





### 3.2.5 Observational needs

The above results are used to motivate recommendations about the spatial and temporal extent to which snow observations should be collected. For example, from Section 3.2.1 and 3.2.2, the usefulness of observations for reducing SWE uncertainty will be higher during the melt season in the high latitudes and western mountainous terrain, whereas having observations in the peak winter is generally more beneficial in the mid-latitudes. Similarly, the timing of snow observations for collecting peak SWE changes with latitude. Finally, the results from Section 3.2.4 point to the need for reliable SWE observations, rather than observations of boundary conditions (such as precipitation) to mitigate the uncertainties in the current state of snow modeling.

### 3.3 Uncertainty Analysis for Different Land Classifications

In this section, we further explore the uncertainty in North American SWE estimates based on different land and snow classifications (described in Section 2.4.2).

### 3.3.1 Uncertainty analysis on different topography

We first evaluate the spatial variability of ensemble mean SWE within each mountain range. In Fig. 7a, box plots #12 and #13 represent the spatial variability of mean SWE for total mountain areas and non-mountain areas, respectively. Total mountain areas are computed by combining the 11 individual mountain domains, and all remaining areas are considered as non-mountain areas. Across the entire continent, the mountain areas show higher spatial variability of SWE and higher median SWE than in non-mountain areas (median SWE: 50.17mm vs. 23.03mm, ~118% higher in mountain areas). Figure 7a highlights that SWE and its spatial variability differ from range to range. For example, most coastal mountain ranges (Coast, Alaska, and Torngat) have higher SWE with greater spatial variability than that of continental ranges (Appalachian, Brooks, Great Basin, Mackenzie, and U.S. Rockies), excluding the Canadian Rockies. Comparisons of SWS in each mountain range (Fig. 7b) show that ~50% of all mountain snow in North America is located in the Coast Range and Canadian Rockies, which is consistent with the findings of Wrzesien et al. (2018).

The variability in the SEUP ensemble spread (i.e., among 12 ensemble members) of SWE and SWS across different mountain ranges is examined in Fig. 7c and 7d. Similar to the spatial variability in SWE (Fig. 7a), the Coast and Alaska ranges have higher uncertainties in SWE among ensemble members, followed by the Cascades, Torngat, and Canadian Rockies. Note that the second highest SWS uncertainty is found in the Canadian Rockies once integrated across the entirety of the mountain range.

To investigate the temporal variability of SWE over different mountain domains, we compared the mean seasonal cycle of SWE and SWS. Figure 8a and 8b show the time series of daily ensemble mean SWE and SWS for each mountain range, averaged for a water year. From both comparisons of SWE and SWS, it can be noted that there is significant variability in the timing of peak (and melt) SWE and SWS across the mountain ranges. The northern mountain ranges (e.g., Alaska, Brooks, Mackenzie, and Torngat) tend to have later dates of peak SWE and SWS, from early April to early May, while peak SWE in





lower latitude mountain ranges occurs between February to March. When exploring the time series of SWE and SWS for each ensemble, we find that JULES simulates non-seasonal snow in the Alaska and Coast mountains (even after the glacier exclusions, not shown), while other LSMs do not. These different estimations are likely due to the different snow physics and
parameterizations used in each LSM (see Table S1). The snow simulated in the summer season could explain the higher spread of SWE seen in Alaska and Coast mountains.

Finally, we use the ensemble-mean seasonal cycle of SWE and SWS to evaluate differences between mountain areas and non-mountain areas of North America. In Fig. 8c and 8d, we find that the daily mean SWE is greater in mountain areas than in non-mountain areas, while the total daily SWS is greater in non-mountain areas than mountain areas. This contrast is due to
the significant difference in total area between the mountain regions and the non-mountain regions: non-mountainous areas cover approximately five times more space than mountainous areas. For total mountain areas, the maximum SWE is 202 mm and the maximum SWS is 616 $km^3$. Alternatively, total non-mountain areas have 79 mm of maximum SWE and 988 $km^3$ of maximum SWS i.e. mountain areas have deeper snow, whereas more snow is stored in non-mountainous areas.

Compared with previous mountain snow studies over North America, the SEUP peak mountain SWS is 1.8 times the
estimate of 342 $km^3$ from the Canadian Sea Ice and Snow Evolution Network (CanSISE) data ensemble of Mudryk et al. (2015) and 0.6 times the estimate of 1,006 $km^3$ in Wrzesien et al. (2018). For non-mountain areas, the SEUP peak SWS is ~1.5 times the estimate of 678 $km^3$ of CanSISE data product. The estimated peak SWS over all of North America from SEUP is 1,604 $km^3$, which is 47.6% more than the previous CanSISE estimate (1,087 $km^3$) and 4.8% less than the Wrzesien et al. (2018) estimate (1,684 $km^3$). When compared with our simulation results, most strikingly, these studies find lower estimation
of SWS even in the non-mountain areas, though additional analysis is needed to determine if this is due to resolution differences or some other influence. The CanSISE SWE estimate is produced using a somewhat similar ensemble mean approach of SEUP, by combining observations and model estimates at 1° spatial resolution. Therefore, the lower estimate of SWS in the CanSISE data product might be explained by their coarser spatial resolution compared to the simulation resolution of this study (i.e., at 5km). Studies such as Broxton et al. (2016) have highlighted the systematic underestimation of SWE from global reanalyses
and continental scale LDASs as a key issue. Previous studies also highlighted the limitations of coarse-resolution models, particularly in capturing snow accumulation in mountain areas, and suggested using a resolution of <10 $km$ (Ikeda et al., 2010; Kapnick & Delworth, 2013; Pavelsky et al., 2011; Wrzesien et al., 2017).

Despite similar identical total North American SWS estimation between SEUP and Wrzesien et al. (2018), there are significant differences in the partitioning between mountain and non-mountain SWS. SEUP estimates that 60% of all
continental snow is located in non-mountains, while Wrzesien et al. (2018) gave an estimation of 60% of all continental snow in mountains. The CanSISE results suggested that ~75% of all continental snow is located in non-mountains, though as noted above, CanSISE estimates may be underestimated due to the coarse modeling resolution, especially in mountain areas. Since Wrzesien et al. (2018) used CanSISE for non-mountain SWS estimates, it is possible that their partitioning of mountain versus non-mountain snow are overestimated. In addition, while SEUP employs ensemble model simulations over an 8-year time
period, Wrzesien et al. (2018) simulated the mountain snowpack using a single regional climate model (i.e., the Weather



Research and Forecasting, WRF version 3.6 (Skamarock et al., 2008), coupled to the Noah-MP3.6 (Niu et al., 2011)) forced by ERA-Interim for a "representative year" (i.e., different single year for each mountain range). This proposed "representative climatology" was used at spatial resolutions of 27 and 9$km$ for the outer and inner domains, respectively. One possible reason for their higher SWS estimates in mountain areas (~63% greater than SEUP) is that their "representative year" had more snow

compared with our average climatology approach over the entire study period, which included low snow (drought) years. Another reason why Wrzesien et al. (2018) had more snow in the mountain is because they used a high-resolution (9km) atmospheric model. The coarser resolution atmospheric models generally do not simulate enough snowfall in the mountains due to their inability to resolve the steepness of the topography (Lundquist et al., 2019). The use of a different glacier mask is another possible explanation for this discrepancy. In addition, it is possible that this single model simulation approach, rather

than using an ensemble method, is biased in mountain areas, though a comparison with the Gravity Recovery and Climate Experiment (GRACE, Syed et al., 2009; Tapley et al., 2004; Wahr et al., 2004) total terrestrial water storage (TWS) anomaly observations showed reasonable results (not shown). Note that any change in TWS from GRACE data is not solely due to snow accumulation or melt. We also compare the variability of SWS among different LSM simulations (not shown) and find that the highest mountain SWS (812 $km^3$) was estimated from Noah-MP3.6 simulations while Wrzesien et al. (2018) showed

the SWS estimate of 1,006 $km^3$ from their simulation of WRF 3.6 using Noah-MP. Note that the Noah-MP3.6 is the most recent and advanced model among SEUP LSMs and has been shown to perform better in previous studies (e.g. Wrzesien et al., 2015).

Overall, the analysis of SWE uncertainty over different topographical regimes confirms that mountain ranges have greater SWE variability among ensemble members than non-mountain regions, likely due to the methods used by the models to resolve

the complex and spatially variable processes over such terrain, and the ability of forcing data to capture orographic effects. These limitations should be addressed through further evaluation of the differences and capabilities of LSMs to simulate of mountain snow, and may also benefit from observational data at high spatio-temporal resolution over such areas. Further, as noted above, there are still significant disagreements in the current understanding of the basic partition of SWE and SWS between mountainous and non-mountainous regions, caused by a variety of factors which are not easy to resolve.

**3.3.2 Uncertainty analysis stratified by snow classes**

The distribution of ensemble mean SWE and SWS (Fig. 9a and 9b) were computed over the entire time period using the snow class definitions shown in Fig. 1. Across the entire continent, the Ice region shows the highest estimate of SWE with the highest spatial variability, followed by Tundra, Taiga, Maritime, Warm Forest, Prairie, and Ephemeral regions. Higher latitudes tend to have higher estimates of SWE and greater spatial variability. Non-seasonal snow was estimated in the Ice region, even

though glaciers were excluded, which may explain the highest SWE and its variability (as discussed earlier in Section 3.3.1). However, the SWS in the Ice regions makes up less than 2.6% of the total over North America. Most strikingly, we find that more than 50% of all continental snow is located in the Tundra region (SWS: 281 $km^3$ with median SWE: 54 mm).



To evaluate SWE uncertainty by different snow regimes, we compare the ensemble spread of mean SWE and SWS for each snow class. Figure 9c and 9d show the spread of mean SWE and SWS for all 12 ensemble members as a function of different snow classes. Both our spatial variability analysis and uncertainty analysis among ensemble members of SWE and SWS estimates provide new insights on the relative importance of different snow classes; the Tundra region has the greatest total SWS and large ensemble spread in those estimates between models; Taiga and Maritime regions also have a significant fraction of the total North American SWS and show high variability in SWE estimates likely due to LSM handling of vegetation impacts, such as canopy interception and sublimation. The SEUP results indicate that SWE estimates in the Tundra region are more consistent between ensemble members, likely because vegetation is sparse there; however, given the large areal extent, accurate SWE estimates are especially critical in estimating total SWS in the Tundra region. Further, we note that the Tundra region is subject to snow erosion and sublimation losses, two processes that the LSMs used in this study do not explicitly simulate. These results point to the need for high accuracy in shallow snow observations that cover large regions, such as Tundra or Prairie, while high spatial resolution in these areas may be less important; the high resolution of SWE observation is more suitable for vegetated areas such as Taiga and Maritime.

### 3.3.3 Influence of Vegetation on SWE Uncertainty

An assessment of snow estimation uncertainty as a function of vegetation is presented in this section. Here we focus primarily on the differences in snow simulations over forested and non-forested areas, since forest snow processes are a model feature that is handled differently between models (Rutter et al., 2009). The forest category includes the evergreen forest, deciduous forest, and mixed forest landcover classes, whereas the non-forest category captures the rest of the landcover categories of Fig. 1d. The spatial variations in ensemble mean SWE as well as the ensemble mean SWS for forested and non-forested areas are shown in Fig. 10a and 10b, respectively. Figure 10a indicates that the non-forested regions have the larger spatial variability than the forested areas. The larger spatial variability in SWE over the non-forested regions is likely explained by the differences in the areal coverage of forests and non-forests (Fig. 1d). The bar plot in Fig. 10b shows that 66% of snow in North America is located in the non-forested regions.

Figure 10c and 10d show the uncertainty in SWE and SWS among the 12 ensembles for forests and non-forests. For both SWE and SWS, the higher spread is seen in the forested regions. This finding is consistent with previous studies that showed the larger spread of snow estimates from model simulations in forested regions (Chen et al., 2014; Essery et al., 2009; Feng et al., 2008; Kim et al., 2019; Rutter et al., 2009). Therefore, these results indicate that future observational efforts should, in part, focus on forested areas and further highlights the need for better understanding the effect of forests on snow simulations.

### 3.4 Uncertainties in the runoff estimation

Since runoff ($R$) estimation, in particular, is significantly influenced by snow evolution, here we examine the impact of uncertainty in SWE estimation on the $R$ estimates and their uncertainty across North America. Similar to Fig. 4, seasonality of $R$ estimates and their uncertainty are evaluated during each winter season and over the entire time period and quantified by





computing the average DOY with the highest ensemble spread and peak $R$. Figure 11 shows the average DOY with the highest

spread in order to identify the: (a) times of high uncertainty in $R$, (b) average DOY with the peak $R$, (c) highest $R$ ensemble

spread, and (d) magnitude of the peak $R$. Variability in the date of the peak $R$ uncertainty and peak $R$ ranges from Jun-Aug in

the high latitudes, whereas at lower latitudes the dates can be outside this range. Similar to the patterns in Fig. 4c and 4d, the

largest spread and peak $R$ amounts are seen along the northern Pacific coastline and in eastern Canada along the northern

Atlantic coastline (excluding the mid-Atlantic and south-eastern U.S.). Figure 11a and 11b indicate that the seasonality in the

highest $R$ spread and highest $R$ values are generally matched. In other words, the largest uncertainty in $R$ occurs at the same

time as the peak $R$, which is different from the patterns shown in Fig. 4 where the largest SWE uncertainty is generally during

the melt season after peak SWE was achieved.

Overall, both Fig. 4 and Fig. 11 show the strong influence of SWE on $R$ over most of North America, and in particular,

during the snow melt season. In order to further examine this, we explore the difference between average DOY of peak SWE

and its spread and average DOY of peak $R$ and its spread. Figure 11e shows this date difference of average DOY of highest

uncertainty (DOY of peak spread in $R$ minus the DOY of peak spread in SWE) and provides a measure of the spatio-temporal

dependence of SWE uncertainty to $R$ uncertainty. Fig. 11f shows the date difference between the average DOY of highest

SWE and highest $R$, which provides a measure of temporal dependence of highest SWE on the highest $R$. If this difference is

negative, it likely indicates that SWE is not a primary driver of runoff. On the other hand, if this difference is positive, it

suggests that SWE has an influence on the runoff regime. The magnitude of this (positive) difference also provides a measure

of the timescale over which they are correlated.

We find, from both figures (11e and 11f), that the times of peak $R$ and uncertainty in peak $R$ occur later in the year than

those of peak SWE and uncertainty in peak SWE over most of the domain. Further, the places where we have the negative

values in both figures are the locations dominated by non-snow $R$ in the lower latitudes. Over the Tundra and Taiga regions,

the differences in the average DOY regimes of SWE and $R$ is about 20-40 days, whereas this lag increases to more than two

months over the Prairie regions. Over the mountainous terrain, $R$ uncertainty is more closely timed with the SWE uncertainty

(~20 days).

This analysis reconfirms that there is generally explicit snow runoff signal during the melt season and increased uncertainty

in $R$ appears related to uncertainty in preceding SWE estimates. Figure 11e and 11f also provide a measure of the spatio-

temporal utility of SWE measurements when considering the objective of improving $R$ estimation. For example, these figures

suggest that SWE estimates approximately 60-80 days prior to the peak flow are likely to provide most utility to $R$ estimation

over the Prairie regions. Since the DOY differences are smaller over the Tundra region, the optimal times for SWE

measurements (20 days prior to the peak flow) are less offset relative to the time of peak $R$. Investigation into the utility of

SWE observations to reduce SWE uncertainty, and thereby runoff uncertainty, will be the focus of future efforts.



## 4 Summary and conclusions

This study employs an ensemble modeling approach to quantify the spatial and temporal uncertainties in SWE over North America, as estimated by operational LSMs and forcing data. Specifically, the study quantifies how uncertainty in SWE varies with key land surface characteristics such as topography, vegetation and snow climate, and evaluates the spatio-temporal

dependence of significant SWE uncertainty on runoff estimation. A primary goal of this study is to establish a baseline assessment of current operational capabilities and identify potential opportunities where improvements or SWE observations could inform both science and application needs.

The SEUP simulated snow estimates are compared against a number of spatially distributed reference snow products, which show a good match over the majority of the modeling domain, with an underestimation over the mountainous regions. The

evaluation metrics provide confirmation that the SEUP ensemble provides a reasonable representation of the snow uncertainty in macroscale snow modeling. Over the entire North American domain, the analysis of the SEUP ensemble indicates that the uncertainty in SWE within this ensemble is driven more by the LSM differences than the choice of forcing data. This suggests that improvements in model physics or increased observations of SWE (ground or remote sensing) rather than improvement in meteorological boundary conditions at this macroscale, are likely to provide more benefit in reducing snow assessment

uncertainty. Though given the underestimation of SWE in mountains by all ensemble members, and high SWE uncertainty found in areas with the deepest snow, particularly the Pacific coastline, higher resolution atmospheric models may be needed to resolve topography and orographic effects in these regions.

Our analysis indicates that there is substantial uncertainty, both SWE and SWS, within forested regions. The Taiga and Maritime regions have a significant fraction of the total North American SWS while also exhibiting high variability in SWE

estimation due to the influence of vegetation. The high spread in SWE and SWS seen over the forested areas suggests the need for improved measurements and modeling of snow in these areas. While these results suggest the need for additional observational constraints to reduce the uncertainty within the models, deep snow and forests also present difficult challenges for remote sensing. These areas continue to be the greatest gaps for global SWE estimation.

The greatest SWS uncertainty is seen in the non-mountainous areas. There are disagreements in the existing literature as to

the relative attribution of snow storage over the mountainous and non-mountainous regions in North America. Though the mountain SWS estimates from SEUP are similar to those generated in prior studies, we conclude that the current partitioning of SWE and SWS between mountainous and non-mountainous areas merits further investigation. Our results provide new insights on the relative importance of the Tundra snow regions where the greatest total SWS is found, and where snowmelt can have important implications on permafrost, arctic ecosystems and global circulation models. Accurate SWE estimates in

shallow snow environments (i.e. tundra and prairie) are critical developing an accurate estimate of global snow partitioning and reducing SWS uncertainty over these regions.

There is significant variability in the seasonality of SWE uncertainty and the uncertainty in peak SWE. At mid-latitudes, the average DOY containing peak SWE and the largest SWE uncertainty occurs in the Dec-Apr time frame. At high latitudes,



particularly in Tundra and Taiga regions, the uncertainty in SWE is largest during May-June, after the peak SWE. These results
suggest that SWE measurements collected during the melt season are likely to provide more benefit in reducing SWE uncertainty at high latitudes and in some western mountainous terrain, whereas observations at (or near) peak SWE accumulation are valuable over the mid-latitudes.

This study also examines the influence of SWE on runoff. The first-order control of SWE on snowmelt runoff over most of North America is highlighted in this study, which points to the importance of improved SWE estimates to inform water
supply and management applications. Overall, this study provides a valuable benchmark on the uncertainties in macroscale snow modeling, which can serve as a guide for prioritizing model improvement needs and developing observational requirements. Additional work is needed to understand the specific drivers of uncertainty within model physics, better characterize the snow storage over mountain and non-mountainous regions, and quantify the regional influence of SWE uncertainty on water availability.

**Data availability**

The Modern-Era Retrospective Analysis for Research and Applications, version 2 (MERRA2; Gelaro et al., 2017; Molod et al., 2015) meteorological data set used in this study is distributed by the NASA Goddard Global Modeling and Assimilation Office (GMAO) (https://gmao.gsfc.nasa.gov/reanalysis/MERRA-2/data_access/). The operational Global Data Assimilation System (GDAS; Derber et al., 1991) data is publicly available from the U.S. National Centers for Environmental Prediction
(NCEP) at https://nomads.ncep.noaa.gov/pub/data/nccf/com/gfs/prod. The European Centre for Medium-Range Weather Forecasts (ECMWF; Molteni et al., 1996) data used in this study is not publicly available and is made available under license (https://www.ecmwf.int/en/forecasts/datasets). The Snow Data Assimilation System (SNODAS; Barrett (2003)) data products, the daily gridded estimates of snow depth and SWE developed by University of Arizona (UA; Zeng et al. (2018)), and the daily, gridded snow depth analysis from the Canadian Meteorological Centre (CMC; Brown and Brasnett (2010)) are available
on National Snow & Ice Data Center (NSIDC)'s data site (https://nsidc.org/data).

**Acknowledgements**

Funding for this work was provided by the NASA Terrestrial Hydrology Program (NNH16ZDA001N). Computing was supported by the resources at the NASA Center for Climate Simulation. NCAR is supported by the National Science Foundation under Cooperative Agreement No. 1852977. Additional support provided by the U.S. Bureau of Reclamation under
Cooperative Agreement Grant Number: R11AC80816 as well as the NASA Advanced Information System Technology Program (80NSSC17K0254).

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

a.


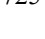


b.

c.



d.

**Figure 1: Snow Ensemble Uncertainty Project (SEUP) domain: (a) domain with terrain elevation. Grey areas indicate the excluded glacier regions, (b) individual mountain domains, (c) individual snow class domains, and (d) land cover classification used in this study.**






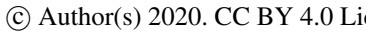

**Figure 2: Maps of average ensemble rank (left column) and Continuous Rank Probability Score (CRPS (mm); right column) from the SEUP ensemble compared to SNODAS (top row), UA (middle row), and CMC (bottom row). SWE is used for SNODAS and UA comparisons, whereas snow depth is used for CMC comparison. Ensemble rank represents the rank of the reference data within the SEUP ensemble, whereas CRPS, which is the extension of mean absolute error to ensemble evaluation, provides a measure of the degree of agreement between the SEUP ensemble and the reference data.**




a.



b.

c.



**Figure 3: (a) Spatial distributions of ensemble mean SWE, (b) the coefficient of variation of ensemble mean SWE, and (c) the range of ensemble mean SWE. The ensemble mean SWE is computed by taking an average of 3 hourly SWE from 12 ensembles over the entire study time period (from 2009 to 2017).**







**Figure 4: Spatial distributions of (a) the peak SWE amount, (b) the highest SWE spread amount, (c) the average day of year (DOY) with the highest ensemble SWE spread, and (d) the difference of average DOY between the highest ensemble SWE spread and the peak SWE (we are only showing/examining places where the DOY difference exist).**




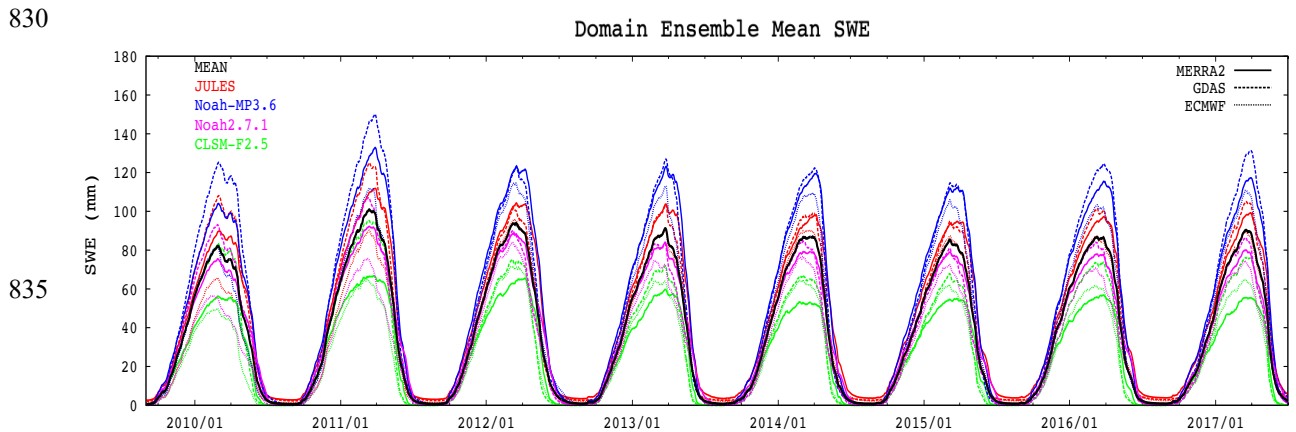

**Figure 5: Time series of domain-averaged mean SWE. Different colors and line style were used to represent each ensemble; a bold**
**black solid line represents the domain-averaged ensemble mean; the units are mm.**

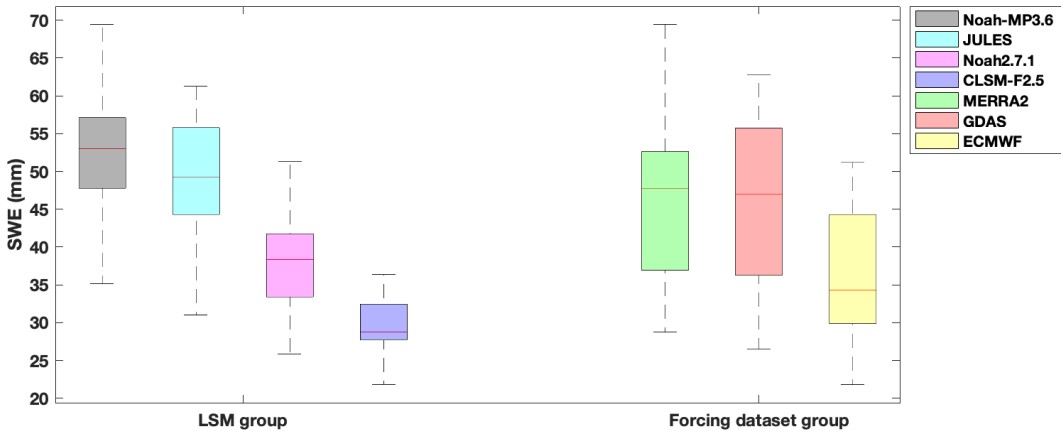

**Figure 6: Distribution of North America mean annual average of SWE (i.e., interannual variability), grouped by the LSMs and**
**forcing datasets (e.g., the box of Noah-MP3.6 represents the distribution of mean SWE, averaged from Noah-MP3.6 runs with all**
**forcing datasets; the box of MERRA2 represents the distribution of mean SWE, averaged from all LSM runs with MERRA2 forcing**
**data). The red line indicates SWE median; top and bottom of box are the 75th and 25th percentiles, and top and bottom of whiskers**
**represent the maximum and minimum SWE without outliers.**




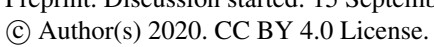

**Figure 7: (a) Spatial variability of ensemble mean SWE (in millimeters) within each mountain range. Red line indicates SWE median; top and bottom of box are the 75th and 25th percentiles and top and bottom of whiskers represent max and min SWE without outliers. (b) Total snow water storage (SWS; in cubic kilometers) within each mountain range, computed from average of ensemble mean SWE over entire time period. The spread of ensembles for (c) domain and time averaged SWE and (d) time averaged SWS for different mountain range.**



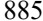








**Figure 8: (a) Climatological SWE (in millimeters) within each mountain range, computed from domain ensemble mean SWE over a water year. (b) Total snow water storage (SWS; in cubic kilometers) climatology within each mountain range, computed from domain ensemble mean SWS over a water year. The mean seasonal cycle of domain averaged SWE (c) and SWS (d) for mountain areas, non-mountain areas, and North America.**





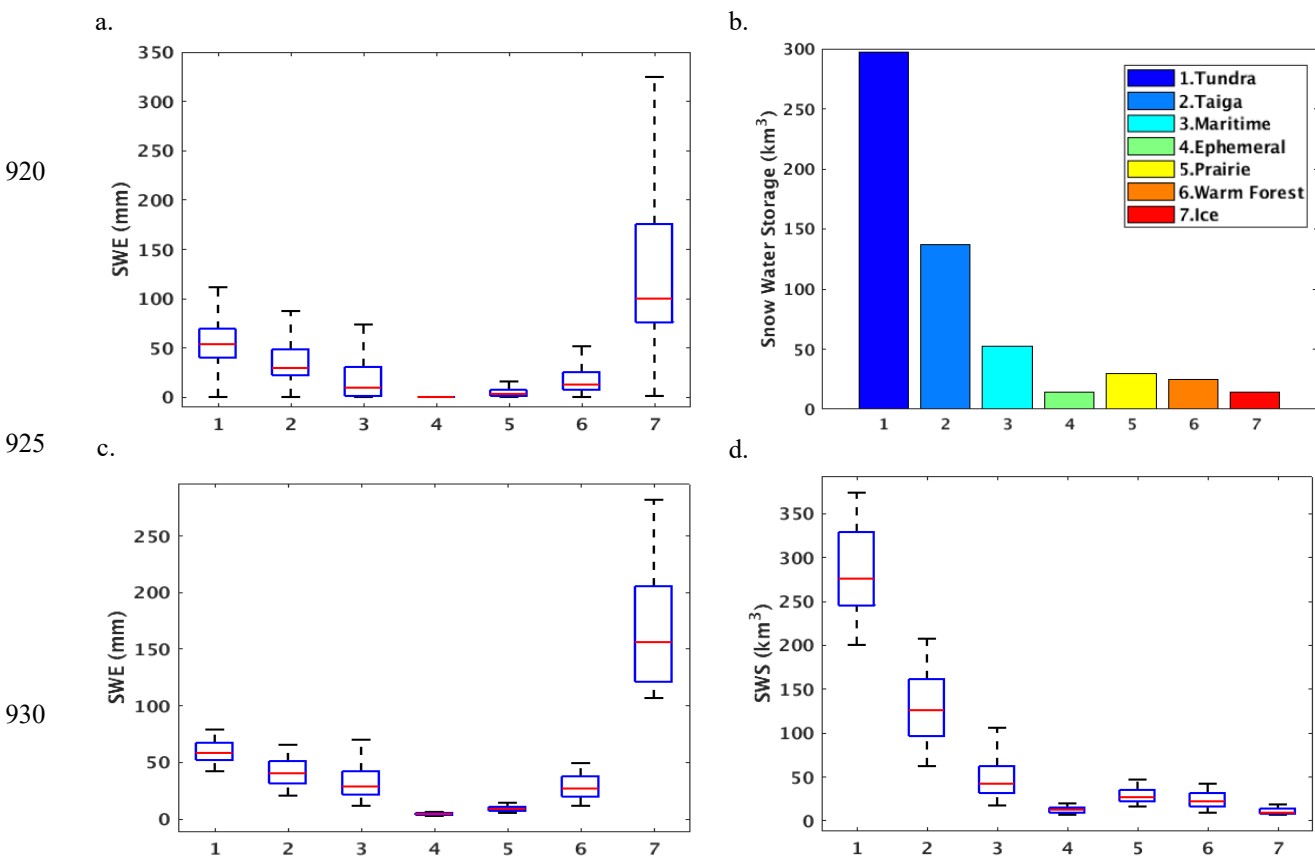

**Figure 9: Same as Figure 7, but for each snow class.**





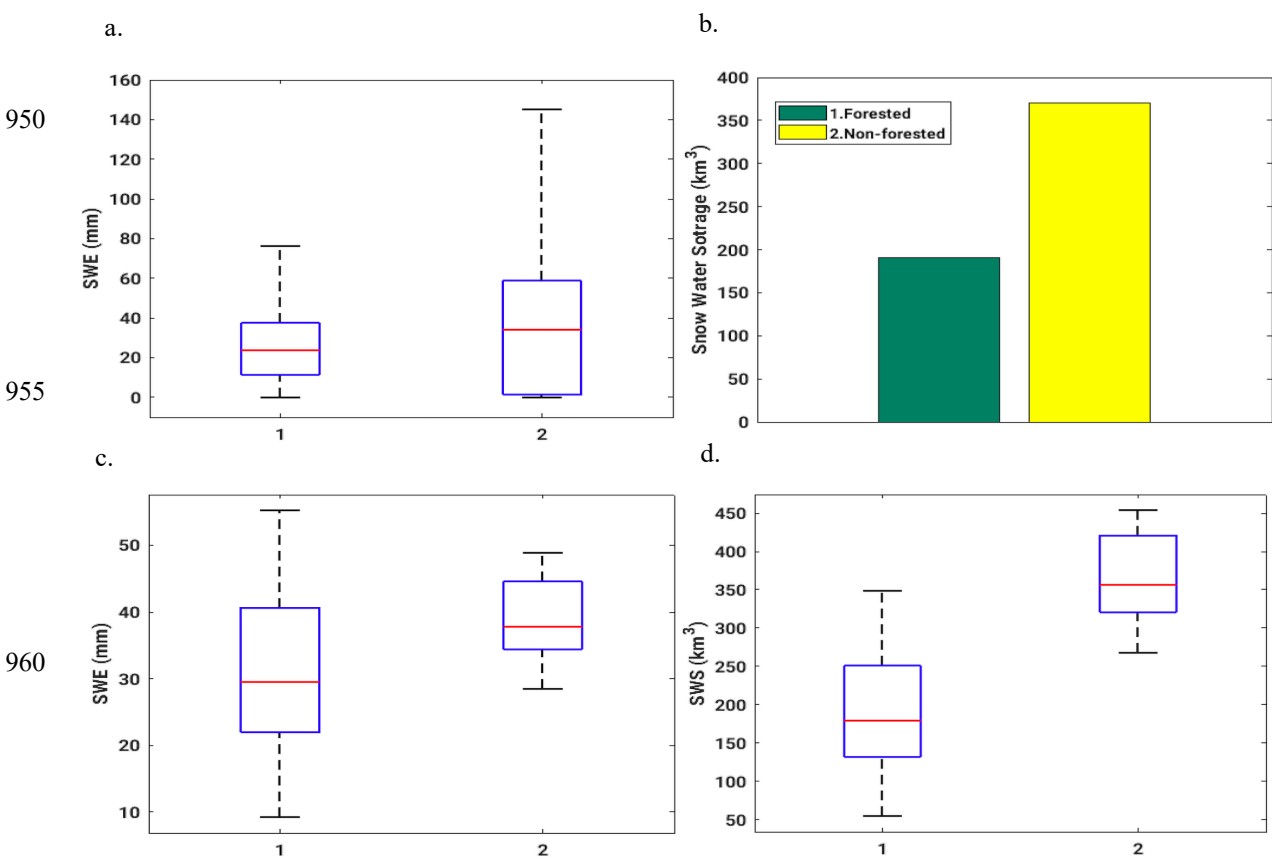

Figure 10: Same as Figure 7, but for forested areas vs non-forested areas.


a.                                              b.

c.                                              d.

e.                                              f.

**Figure 11: Spatial distributions of (a) the average day of year (DOY) with the highest ensemble $R$ spread, (b) the average DOY with the peak $R$, (c) the highest $R$ spread amount, (d) the peak $R$ amount, (e) the difference of average DOY between the highest $R$ spread and the highest SWE spread, and (f) the difference of average DOY between the peak $R$ and the peak SWE.**
