# Peer review of "Snow Ensemble Uncertainty Project (SEUP): Quantification of snow water equivalent uncertainty across North America via ensemble land surface modeling"

_The Cryosphere, 2020_

## Referee Comment (RC1) · Richard L.H. Essery (Referee) · 18 Oct 2020

Kim et al. quantify uncertainty in estimates of snow storage by a set of operational models and forcing datasets. This is a topic of considerable interest, given the importance of snow storage for water resources and the difficulty of estimating it by remote sensing. The paper is interesting and generally well written; my comments focus on clarifying the methodology.

119

[Figure]

Why was the three-layer snow scheme in JULES, described by Best et al. (2011) and operational at the Met Office, not used?

122

"not tuned in this study to assess current configurations" is ambiguous. I assume that "to assess current configurations, parameters were not tuned" is what is intended.

130

State the original spatial and temporal resolutions of the forcing datasets.

132

Doesn't SRTM extend only to 60N? How is the downscaling achieved up to 71.875N?

134

How are relative humidity, wind and longwave radiation downscaled? I don't think that this is described in Cosgrove et al. (2003).

137

It is worth noting that Kumar et al. (2013) concluded that topographic adjustments to radiation should be included in models with resolutions finer than 16 km, but the adjustments are likely to be small at 5 km resolution.

154

Should this be "If the observation is more than 10% higher than the highest ensemble member, then the rank is set to 13"?

161

Conventionally, an integral would not have the "X=" in its limits. "P0 represents the observations occurrence" is not clear – it is a step function at the observed value.

184

To be clear, CMC includes an estimate of SWE, but only snow depth observations are used in the CMC analysis.

222

Give some context for what can be regarded as a "low" value of CRPS.

243

Some information should be given (earlier, or refer to S1) on how rain/snow partitioning schemes differ between the LSMs. Three of the four are identical.

284

Raleigh et al. (2016) and Guenther et al. (2019) added uncertainty to single forcing datasets, rather than using multiple forcing datasets as here.

381

Is SWS averaged over the entire time period meaningful for comparing different snow classes, when it conflates the amount and duration of snow cover? How about looking at average annual maximum SWS?

456 For "c urrent operational capabilities", note that several countries now have operational limited area numerical weather prediction models with spatial resolutions on the order of 1 km.

Figure 2 caption

Average CRPS.

Move the information "13 is more snow than all ensembles and 0 is less snow than all ensembles" from the figure to the caption

Figure 6

Does "without outliers" mean that outliers are omitted from the diagram (in which case,

how is an outlier defined?) or that the ends of the whiskers are the SWE range?

There is room to put the LSM and forcing dataset labels on the figure axis, removing the need for the reader to relate the figure legend to the boxes and removing the need for colour.

If I understand correctly, five statistics (minimum, 25th percentile, median, 75th percentile and maximum mean annual average of SWE) have been calculated from just three data points for each LSM (mean annual average SWE for three different forcing datasets). Why not just show the points?

Figures 1, 3, 4 and 11

Rainbow colour scales are widely deprecated.

If latitudes and longitudes are not going to be marked, remove the redundant grid lines.

For the figures that use a divergent colour scale centred on white, the coastline would be a nice addition.

Figures 7, 9 and 10

All of the colours in these figures are redundant.

Minor corrections:

49

"National Academies"

203

"the influence of vegetation on SWE uncertainty"

255

Remove redundant words: "ranges from December-April in the lower latitudes to May-June in the high latitudes"

[Figure]

278

"across the forcing datasets when driving a common LSM"

260

SW spread is Fig 4b.

323

"between February and March"

324

"JULES simulates permanent snow"

341

"a somewhat similar ensemble mean approach to SEUP"

348

Similar, not identical.

407

"non-forested regions have larger spatial variability"

417

"Since runoff (R), in particular, is significantly influenced by snow evolution"

480

"critical to developing"

---

## Referee Comment (RC2) · J. Ignacio López-Moreno (Referee) · 20 Oct 2020

The paper analyses the uncertainty of snow simulations by different combinations of global Land Surface models (LSMs) and forcing datasets (FDs) over North America. The paper is interesting and provides useful insights about the applicability of climate models to have reasonable estimates of snow storage for large areas. The paper is well written and structured, and may be of interest for a wide variety of readers. Despite is not its main objective of the paper I miss a bit more of discussion about the causes of the detected uncertainties, and to present some comparison of the individual members

of the ensemble with the snow datasets used as references. Such information could offer clues about the origin of some uncertainties (i.e. if uncertainties are more related to the parametrization of the snow processes, or to difficulties of the models to reproduce the driving variables of their snow energy balance (precipitation, temperature. . .); or even an individual analysis of uncertainty may open the possibility to consider reducing the number of members of the ensemble if some of them shows a clear systematic bias with the reference datasets. At some point is mentioned that "combining a variety of model estimates and allowing the individual model errors to cancel each other out (Xia et al., 2012)". I think this may be true when errors are random or the causes behind the errors are not well identified. However, if some member fails systematically because the snow parametrization is too simplistic or is using more limited observations than others, it is possible preferably to leave out such members from the final ensemble. In addition if some of the forcing data uses observations that are not available, or their density is more reduced, out of the domain of this study (North America) and they provide better results than others, it could be discussed in the manuscript as it has implications when used in other regions of the world.

Specific comments - It is a bit surprising to me finding in the areas with deeper snowpack the largest uncertainties, since simulating shallow snowpacks is often more challenging than deeper ones. I would like to hear the hypothesis of the authors about this result. Are the climate (i.e. precipitation) in these areas more difficult to be simulated? Is it more/less affected to uncertainties in snow-rain separation? Same for different uncertainty between flat areas and mountains (Forcing datasets have less observations in mountains? The uncertainty in mountains is associated to the downscaling technique, or again the reproducibility of the climate there? If the individual members are compared to the reference snow datasets, the complexity of the snow-vegetation parametrization could be used to explain results when forested and non-forested areas are compared? The use of 1 or several layers for snowpack could be also identified as potential source of uncertainty shown in the study. - Line 73: 5 km instead of 5km -Line 207: "Section 3.1 compares the ensemble with observations derived from data

assimilation techniques" I would not call them observations, perhaps is better using comparison with the reference snow datasets Section 3.2.2. Not sure if seasonal variability is the most accurate tittle for the subsection, what about: "Timing of annual peak SWE…" - Figure 5 and 6: £Is it possible to add boxplots with the values for the snow datasets used as references? It can give a good indication about some specific LSM or forcing dataset clearly biased from "reality". -It would be also good to show a figure as 5 (probably as supplementary) but specifically for areas where the annual behavior of snow patterns is known to be well contrasted (often with opposite anomalies; i.e. Cascades and Rockies US) because the average over the entire domain smooths the differences and lead to very little interannual variability. - The subsection Section 3.2.5 directly links higher uncertainty with lack of observations. I am not sure if this statement is sound because the reasons of the uncertainties are not well identified along the manuscript. -Line 368: referring to GRACE: "total terrestrial water storage (TWS) anomaly observations showed reasonable results (not shown)." I do not understand what authors really mean. - Caption of Figure 8 says "Rockies Canaidan" - Section 3.4 is interesting and indeed the topic could be a new manuscript itself. However, as it is presented I have the feeling that many of the conclusions derived from this section are not fully supported needing deeper analyses. Authors may consider remove this section that could compensate to develop more other sections (i.e. assessment of uncertainty of the individual members of the ensemble).

J. Ignacio López-Moreno

---

## Referee Comment (RC3) · Anonymous Referee #3 · 22 Oct 2020

The authors present a highly interesting and relevant study assessing snow modelling uncertainties across the North American region with (among others) the goal of providing information about global snow observation needs. The study comprises four different land surface models and three forcing data sets, resulting in an ensemble of twelve members. The results show that the uncertainty represented by the ensemble spread varies across the study domain, with for instance high uncertainty in the simulation results for mountainous and forested regions. The authors conclude that for these regions high-resolution observations are needed to capture the high spatial variability in snow water equivalent. Overall, the article is very well written, easy to follow and from the technical perspective ready for publication in my opinion. Nonetheless, the current manuscript only provide rather shallow information about what snow observations are required in order to reduce the uncertainty seen in the simulation results. What variables should be measured and on which spatial and temporal resolution? For example, the need for high-resolution observations in mountains to capture the high spatial snow water equivalent variability has been recognized a long time ago. However, for very large domains, we still lack observations with sufficient quality as well as high enough spatial and temporal resolution, and this may not change soon either. Providing some more details, foremost more quantitative, about the observational needs to constrain the model uncertainties would make the paper even more interesting to read.

―――――――――――――――――――――

---

## Author Comment (AC1) · 10 Dec 2020

Dear Dr. Essery,

Thank you for your constructive comments. Based on the feedback, we have made several changes and improvements to the manuscript. Below we have provided a response to each of the comments provided.

Richard L.H. Essery (Referee1)'s comments:

[Figure]

Kim et al. quantify uncertainty in estimates of snow storage by a set of operational models and forcing datasets. This is a topic of considerable interest, given the importance of snow storage for water resources and the difficulty of estimating it by remote sensing. The paper is interesting and generally well written; my comments focus on clarifying the methodology.

119 Why was the three-layer snow scheme in JULES, described by Best et al. (2011) and operational at the Met Office, not used?

The reviewer is correct that currently, the Met Office uses a three-layer snow scheme which was deployed in 2018, post the modeling setups were devised for this study.

We have revised the text to clarify in lines 124-125 on page 4, "Note that the UKMO currently uses a three-layer scheme in JULES, which wasn't available in NASA LIS at the time this study was devised."

122 "not tuned in this study to assess current configurations" is ambiguous. I assume that "to assess current configurations, parameters were not tuned" is what is intended.

We have revised this sentence in lines 125-126 on page 4 as: "In order to assess current configurations, initial model conditions and model parameters used in the operational set-up were not tuned in this study."

130 State the original spatial and temporal resolutions of the forcing datasets.

We have added the following sentences in the main manuscript and S2.

Main manuscript (in lines 133-134 on page 5): "Original spatial and temporal resolutions for these datasets are described in Section S2".

Section S2.1 (in lines 107-108 on page 4 in supporting information): "MERRA2 has a native spatial resolution of $0.5°$ latitude by $0.625°$ longitude (roughly 50 km) and hourly temporal resolution."

Section S2.2 (in lines 113-115 on page 4 in supporting information): "The GDAS model

grids have been upgraded from roughly 80 km (since 2000), ∼60 km (Oct. 2002), ∼38 km (Jun. 2005), 27 km (Jul. 2010), to ∼13 km (January 2015). The temporal resolution is 3-hour."

Section S2.3 (in lines 119-122, on page 4 in supporting information): "In this study, the operational real-time data from the ECMWF-Integrated Forecast System (IFS) are used; the meteorological fields are provided on a 0.25-degree grid (roughly 25 km) at a 3-hour interval, and generated by assimilating available atmospheric observations every 12 hours into a forecast model with surface meteorological fields (e.g., precipitation and radiation), which are diagnosed from the model output (Dee et al., 2011; Flemming et al., 2015)."

132 Doesn't SRTM extend only to 60N? How is the downscaling achieved up to 71.875N?

The reviewer is correct that SRTM extends only to 60N. We use the GTOPO30 dataset for regions north of 60N. We have modified the text in lines 136-139 on pages 5 as follows:

"Meteorological inputs of near surface air temperature, relative humidity, surface pressure, and downward longwave radiation are downscaled by applying a lapse-rate and hypsometric adjustments using the 5 km Shuttle Radar Topography Mission (SRTM; between 60N and 60S) and the USGS Global 30 arc second elevation (GTOPO30; north of 60N) elevation datasets."

134 How are relative humidity, wind and longwave radiation downscaled? I don't think that this is described in Cosgrove et al. (2003).

The wind fields are not adjusted for topographic differences. This was erroneously stated in the manuscript and we have removed it. The lapse-rate correction of near surface air temperature, relative humidity, surface pressure, and downward longwave radiation is described in Cosgrove et al. (2003). The manuscript also had the incorrect

"Cosgrove et al." reference. It has been corrected in line 140 on page 5 to the following: Cosgrove, B. A., et al. (2003), Real‐time and retrospective forcing in the North American Land Data Assimilation System (NLDAS) project, J. Geophys. Res., 108, 8842, doi:10.1029/2002JD003118, D22.

137 It is worth noting that Kumar et al. (2013) concluded that topographic adjustments to radiation should be included in models with resolutions finer than 16 km, but the adjustments are likely to be small at 5 km resolution.

Thanks for pointing this out. We have added this sentence in lines 143-146 on page 5 as follows:

"Kumar et al. (2013) demonstrated that these adjustments are particularly important for improving snow simulations over midlatitude domains in regions of complex topography and concluded that these adjustments should be included in models with resolutions finer than 16 km, but the adjustments are likely to be small at 5 km resolution."

154 Should this be "If the observation is more than 10% higher than the highest ensemble member, then the rank is set to 13"?

Thanks for this correction. We've changed this sentence in lines 160-161 on page 5, "If the observation is more than 10% higher than the highest ensemble member, then the rank is set to 13."

161 Conventionally, an integral would not have the "X=" in its limits. "P0 represents the observations occurrence" is not clear – it is a step function at the observed value.

We have replaced the equation 1 and its description as follows (in lines 168-170 on page 6), please find the attached file to see this response):

184 To be clear, CMC includes an estimate of SWE, but only snow depth observations are used in the CMC analysis.

We have replaced this sentence in lines 191-192 on page 6 as follows:

"Despite providing an estimate of SWE, in this analysis, we evaluate the CMC modeled snow depth fields since the CMC only uses snow depth observations in its analysis."

222 Give some context for what can be regarded as a "low" value of CRPS.

As noted in the text, CRPS can be considered analogous to mean absolute error, but for ensembles. It is hard to define a single value to define a "low" value of CRPS. We have modified the text in lines 227-228 on page 8 as follows to quantify what we mean by "low":

"Over most of the domain, including the northeast/Midwest U.S. and high plains, the CRPS values are low (0-100 mm), where a low (good) score indicates a small ensemble spread that agrees with SNODAS and UA data."

243 Some information should be given (earlier, or refer to S1) on how rain/snow partitioning schemes differ between the LSMs. Three of the four are identical.

We have added this information in lines 248-251 on page 8 as follows:

"These highly complex terrains have relatively high snowfall precipitation, and the large spread is partially due to different rain/snow partitioning schemes in each LSM. While Noah2.7.1, JULES, and CLSMF-2.5 use a simple temperature threshold of 0oC to distinguish rainfall and snowfall precipitation, Noah-MP3.6 includes a transition temperature range described in Jordan (1991) (see Table S1)."

We've also added Jordan's (1991) scheme for the Noah-MP3.6's precipitation partitioning method in the Table S1 (please find the attached file to see this response).

284 Raleigh et al. (2016) and Guenther et al. (2019) added uncertainty to single forcing datasets, rather than using multiple forcing datasets as here.

We have revised this sentence in lines 297-299 on page 10 as follows:

"For example, Raleigh et al. (2016) and Günther et al. (2019) showed the forcing data to be the primary driver of SWE uncertainty in their study, which used a single forcing

dataset with added uncertainty and focused on a limited number of relatively small sites mostly in mountainous terrains."

381 Is SWS averaged over the entire time period meaningful for comparing different snow classes, when it conflates the amount and duration of snow cover? How about looking at average annual maximum SWS?

Thanks for this comment. We agree that looking at the average annual maximum SWS is another interesting way. However, using annual maximum SWS in this study is not ideal because we're evaluating over a large spatial extent. Choosing the date of the annual maximum SWS is also challenging. The date of the annual maximum SWS would be different between snow classes and these are also different from the date of domain maximum SWS.

456 For "current operational capabilities", note that several countries now have operational limited area numerical weather prediction models with spatial resolutions on the order of 1 km.

We have revised this sentence in lines 469-471 on page 15 as follows:

"A primary goal of this study is to establish a baseline assessment of current global- or continental-scale operational capabilities and identify potential opportunities where improvements or SWE observations could inform both science and application needs."

Figure 2 caption

Average CRPS. Move the information "13 is more snow than all ensembles and 0 is less snow than all ensembles" from the figure to the caption

We have corrected this caption.

Figure 6 Does "without outliers" mean that outliers are omitted from the diagram (in which case, how is an outlier defined?) or that the ends of the whiskers are the SWE range?

The outlier is defined as a value that is more than 1.5 times the interquartile range away from the top or bottom of the box.

"without outliers" has been corrected in Figures 6 and 7 to "with outliers (defined as more than 1.5 times the interquartile range (between 25% and 75%)) omitted".

There is room to put the LSM and forcing dataset labels on the figure axis, removing the need for the reader to relate the figure legend to the boxes and removing the need for colour.

The plot has been modified.

If I understand correctly, five statistics (minimum, 25th percentile, median, 75th percentile and maximum mean annual average of SWE) have been calculated from just three data points for each LSM (mean annual average SWE for three different forcing datasets). Why not just show the points?

These statistics were calculated using the annual average of SWE for each year (i.e., interannual variability). For the LSM group, we used 8 annual averages of SWE (from 2009 to 2017) for three different forcing datasets (8*3). For the forcing dataset group analysis, 8 annual averages of SWE for four different (8*4) LSMs were used.

We have added these details in Figure 6 caption as follows: "Distribution of North America mean annual average of SWE (i.e., interannual variability), grouped by the LSMs and forcing datasets (e.g., the box of Noah-MP3.6 represents the distribution of mean SWE, averaged from Noah-MP3.6 runs with all forcing datasets; the box of MERRA2 represents the distribution of mean SWE, averaged from all LSM runs with MERRA2 forcing data). For the LSM group, we used 8 annual averages of SWE (from 2009 to 2017) for three different forcing datasets (total of 8*3). For the forcing dataset group, 8 annual averages of SWE for four different LSMs (total of 8*4) were used. The red line indicates SWE median; top and bottom of box are the 75th and 25th percentiles, and top and bottom of whiskers represent the maximum and minimum

SWE with outliers (defined as more than 1.5 times the interquartile range (between 25% and 75%)) omitted."

Figures 1, 3, 4 and 11 Rainbow colour scales are widely deprecated. If latitudes and longitudes are not going to be marked, remove the redundant grid lines. For the figures that use a divergent colour scale centered on white, the coastline would be a nice addition.

Thank you for the suggestion. Figures 1,3,4 and 11 have been revised.

Figures 7, 9 and 10 All of the colours in these figures are redundant.

Figures 7,9 and 10 have been revised.

References

Dee, D. P., Uppala, S. M., Simmons, A. J., Berrisford, P., Poli, P., Kobayashi, S., Andrae, U., Balmaseda, M. A., Balsamo, G., Bauer, P., Bechtold, P., Beljaars, A. C. M., van de Berg, L., Bidlot, J., Bormann, N., Delsol, C., Dragani, R., Fuentes, M., Geer, A. J., Haimberger, L., Healy, S. B., Hersbach, H., Hólm, E. V., Isaksen, L., Kållberg, P., Köhler, M., Matricardi, M., Mcnally, A. P., Monge-Sanz, B. M., Morcrette, J. J., Park, B. K., Peubey, C., de Rosnay, P., Tavolato, C., Thépaut, J. N. and Vitart, F.: The ERA-Interim reanalysis: Configuration and performance of the data assimilation system, Quarterly Journal of the Royal Meteorological Society, doi:10.1002/qj.828, 2011.

Essery, R., Rutter, N., Pomeroy, J., Baxter, R., Stahli, M., Gustafsson, D., Barr, A., Bartlett, P. and Elder, K.: SnowMIP2: An evalution of forest snow process simulation, Bulletin of the American Meteorological Society, doi:10.1175/2009BAMS2629.1, 2009.

Flemming, J., Huijnen, V., Arteta, J., Bechtold, P., Beljaars, A., Blechschmidt, A. M., Diamantakis, M., Engelen, R. J., Gaudel, A., Inness, A., Jones, L., Josse, B., Katragkou, E., Marecal, V., Peuch, V. H., Richter, A., Schultz, M. G., Stein, O. and Tsikerdekis, A.: Tropospheric chemistry in the integrated forecasting system of ECMWF, Geoscientific Model Development, doi:10.5194/gmd-8-975-2015, 2015.

Günther, D., Marke, T., Essery, R. and Strasser, U.: Uncertainties in snowpack simulations assessing the impact of model structure, parameter choice, and forcing data error on point-scale energy balance snow model performance, Water Resources Research, doi:10.1029/2018WR023403, 2019.

Jordan, R.: A one-dimensional temperature model for a snow cover: Technical documentation for SNTHERM.89., 1991.

Raleigh, M. S., Livneh, B., Lapo, K. and Lundquist, J. D.: How does availability of meteorological forcing data impact physically based snowpack simulations?, Journal of Hydrometeorology, 17(1), 99–120, doi:10.1175/JHM-D-14-0235.1, 2016.

Rutter, N., Essery, R., Pomeroy, J., Altimir, N., Andreadis, K., Baker, I., Barr, A., Bartlett, P., Boone, A., Deng, H., Douville, H., Dutra, E., Elder, K., Ellis, C., Feng, X., Gelfan, A., Goodbody, A., Gusev, Y., Gustafsson, D., HellströM, R., Hirabayashi, Y., Hirota, T., Jonas, T., Koren, V., Kuragina, A., Lettenmaier, D., Li, W.-P., Luce, C., Martin, E., Nasonova, O., Pumpanen, J., Pyles, R. D., Samuelsson, P., Sandells, M., SchäDler, G., Shmakin, A., Smirnova, T. G., StäHli, M., StöCkli, R., Strasser, U., Su, H., Suzuki, K., Takata, K., Tanaka, K., Thompson, E., Vesala, T., Viterbo, P., Wiltshire, A., Xia, K., Xue, Y. and Yamazaki, T.: Evaluation of forest snow processes models (SnowMIP2), Journal of Geophysical Research: Atmospheres, 114(D), D06111, doi:10.1029/2008JD011063, 2009.

Please also note the supplement to this comment:
https://tc.copernicus.org/preprints/tc-2020-248/tc-2020-248-AC1-supplement.pdf
* * *
[Figure]

a.

Elevation (m)

b.

1.Alaska
2.Appalachian
3.Brooks
4.Cascades
5.Coast
6.Great Basin
7.Mackenzie
8.Rockies,Canaidan
9.Rockies,USA
10.Sierra Nevada
11.Torngat

c.

1.Tundra
2.Taiga
3.Maritime
4.Ephemeral
5.Prairie
6.Warm Forest
7.Ice

d.

1. Forested
2. Non-forested

**Figure 1: Snow Ensemble Uncertainty Project (SEUP) domain: (a) domain with terrain elevation. Grey areas indicate the excluded glacier regions, (b) individual mountain domains, (c) individual snow class domains, and (d) land cover classification used in this study.**

**Fig. 1.**

[Figure]

[Figure]

**Figure 2:** Maps of average ensemble rank (left column) and average Continuous Rank Probability Score (CRPS (mm); right column) from the SEUP ensemble compared to SNODAS (top row), UA (middle row), and CMC (bottom row). SWE is used for SNODAS and UA comparisons, whereas snow depth is used for CMC comparison. Ensemble rank represents the rank of the reference data within the SEUP ensemble. Rank 13 represents more snow than all ensembles and rank 0 is less snow than all ensembles. CRPS, which is the extension of mean absolute error to ensemble evaluation, provides a measure of the degree of agreement between the SEUP ensemble and the reference data.

**Fig. 2.**

* * *
**Figure 3:** (a) Spatial distributions of ensemble mean SWE, (b) the coefficient of variation of ensemble mean SWE, and (c) the range of ensemble mean SWE. The ensemble mean SWE is computed by taking an average of 3 hourly SWE from 12 ensembles over the entire study time period (from 2009 to 2017).

**Fig. 3.**

[Figure]

[Figure]

a.

b.

c.

d.

Figure 4: Spatial distributions of (a) the peak SWE amount, (b) the highest SWE spread amount, (c) the average day of year (DOY) with the highest ensemble SWE spread, and (d) the difference of average DOY between the highest ensemble SWE spread and the peak SWE (we are only showing/examining places where the DOY difference exist).

Fig. 4.

[Figure]

Figure 6: Distribution of North America mean annual average of SWE (i.e., interannual variability), grouped by the LSMs and forcing datasets (e.g., the box of Noah-MP3.6 represents the distribution of mean SWE, averaged from Noah-MP3.6 runs with all forcing datasets; the box of MERRA2 represents the distribution of mean SWE, averaged from all LSM runs with MERRA2 forcing data). For the LSM group, we used 8 annual averages of SWE (from 2009 to 2017) for three different forcing datasets (total of 8*3). For the forcing dataset group, 8 annual averages of SWE for four different LSMs (total of 8*4) were used. The red line indicates SWE median; top and bottom of box are the 75th and 25th percentiles, and top and bottom of whiskers represent the maximum and minimum SWE with outliers (defined as more than 1.5 times the interquartile range (between 25% and 75%)) omitted.

**Fig. 5.**

[Figure]

Figure 7: (a) Spatial variability of ensemble mean SWE (in millimeters) within each mountain range. Red line indicates SWE median; top and bottom of box are the 75th and 25th percentiles and top and bottom of whiskers represent max and min SWE with outliers (defined as more than 1.5 times the interquartile range (between 25% and 75%)) omitted. (b) Total snow water storage (SWS; in cubic kilometers) within each mountain range, computed from average of ensemble mean SWE over entire time period. The spread of ensembles for (c) domain and time averaged SWE and (d) time averaged SWS for different mountain range.

**Fig. 6.**

a.

[Figure]

b.

1. Tundra
2. Taiga
3. Maritime
4. Ephemeral
5. Prairie
6. Warm Forest
7. Ice

c.

d.

Figure 9: Same as Figure 7, but for each snow class.

**Fig. 7.**

a.

[Figure]

1. Forested
2. Non-forested

b.

c.

d.

Figure 10: Same as Figure 7, but for forested areas vs non-forested areas.

Fig. 8.

[Figure]

[Figure]

Figure 11: Spatial distributions of (a) the average day of year (DOY) with the highest ensemble *R* spread, (b) the average DOY with the peak *R*, (c) the highest *R* spread amount, (d) the peak *R* amount, (e) the difference of average DOY between the highest *R* spread and the highest SWE spread, and (f) the difference of average DOY between the peak *R* and the peak SWE.

**Fig. 9.**

**Supplement:**

161
Conventionally, an integral would not have the "X=" in its limits. "P0 represents the observations occurrence" is not clear – it is a step function at the observed value.

We have replaced the equation 1 and its description as follows (in lines 168-170 on page 6):

"$CRPS = \int_{-\infty}^{+\infty}(P_m - P_o)^2 dx$ ,

where $P_m$ represents the cumulative distribution function (CDF) of the model and $P_o$ represents the Heaviside step function at the observed value."

243
Some information should be given (earlier, or refer to S1) on how rain/snow partitioning schemes differ between the LSMs. Three of the four are identical.

We have added this information in lines 248-251 on page 8 as follows:

"These highly complex terrains have relatively high snowfall precipitation, and the large spread is partially due to different rain/snow partitioning schemes in each LSM. While Noah2.7.1, JULES and CLSMF-2.5 use a simple temperature threshold of 0°C to distinguish rainfall and snowfall precipitation, Noah-MP3.6 includes a transition temperature range described in Jordan (1991) (see Table S1)."

We've also added Jordan's (1991) scheme for the Noah-MP3.6's precipitation partitioning method in the Table S1.

"Snow for $T_{air}<0.5°C$, rain for $T_{air}>2.5°C$,
Snowfall fraction=0.6: $2.0°C< T_{air}\leq2.5°C$,
Linear rain/snow transition: $0.5°C< T_{air}\leq2.0°C$"

---

## Author Comment (AC2) · 10 Dec 2020

Dear Dr. López-Moreno,

Thank you for your constructive feedback. Below we have provided a response to each of the comments provided by the referee.

J. Ignacio López-Moreno (Referee2)'s comments: The paper analyses the uncertainty of snow simulations by different combinations of global Land Surface models (LSMs) and forcing datasets (FDs) over North America. The paper is interesting and provides

useful insights about the applicability of climate models to have reasonable estimates of snow storage for large areas. The paper is well written and structured, and may be of interest for a wide variety of readers. Despite is not its main objective of the paper I miss a bit more of discussion about the causes of the detected uncertainties, and to present some comparison of the individual members of the ensemble with the snow datasets used as references. Such information could offer clues about the origin of some uncertainties (i.e. if uncertainties are more related to the parametrization of the snow processes, or to difficulties of the models to reproduce the driving variables of their snow energy balance (precipitation, temperature. . .); or even an individual analysis of uncertainty may open the possibility to consider reducing the number of members of the ensemble if some of them shows a clear systematic bias with the reference datasets. At some point is mentioned that "combining a variety of model estimates and allowing the individual model errors to cancel each other out (Xia et al., 2012)". I think this may be true when errors are random or the causes behind the errors are not well identified. However, if some member fails systematically because the snow parametrization is too simplistic or is using more limited observations than others, it is possible preferably to leave out such members from the final ensemble. In addition if some of the forcing data uses observations that are not available, or their density is more reduced, out of the domain of this study (North America) and they provide better results than others, it could be discussed in the manuscript as it has implications when used in other regions of the world.

Thank you for bringing out these issues in the interactive discussion. First, we want to emphasize that the main objective of this paper is to establish an important baseline over the continental scales to characterize current capabilities and inform global snow observational requirements by using a range of forcing products and commonly-used operational models. Given this set-up, we are defining uncertainty as the ensemble spread between these, knowing that the entire set might miss the truth, but assuming that times/places with a higher spread do validly represent greater uncertainties at times/places to focus more measurements or efforts-to-improve understanding. In

addition, we acknowledge that some of these models are not state of the art (e.g., we know there are publications saying that CLSM-F2.5 isn't really representative of state-of-the-art in snow modeling) and some of these are not independent samples (e.g. the models have similar underlying parameterizations), but these are representative of data that are currently available and that people might use for their own research, given that these models and configurations are featured in operational systems.

To acknowledge your comment, we added the following text on lines 108-112 on page 4 as follows: "Note that some of these models do not necessarily represent the state-of-the-art approaches for snow modeling and their underlying parameterizations may share a similar legacy in terms of code development. Despite these limitations, however, these models and their versions are representative of systems that provide publicly available snow estimates over continental and global scales."

We are also explicitly looking at two sources of spread – the met data and the model structure (again, from an operational suite), because the literature diverges about which is a greater source of uncertainty. Thus, the SEUP effort also attempts to differentiate from the earlier snow model intercomparison experiments (Essery et al. 2009; Rutter et al. 2009), which assessed errors associated with model structure and parameterizations. In this paper, we are neglecting other sources of uncertainty such as the choice of parameters or resolution of the topography. While understanding the causes of the detected uncertainties and assessment of uncertainty of the individual members of the ensemble is important, we consider it to be outside the scope of this study and leave it for future follow-on efforts.

We have added the following clarification in lines 506-507 on page 16 as follows: "We acknowledge that other sources of uncertainty such as the choice of model parameters and spatial resolution of topographic features are not examined here."

For forcing data concerns in other regions, we briefly discussed this in lines 299-301 on page 10 referred to Yoon et al. (2019)'s study: "Similarly, Yoon et al. (2019) recently showed that the forcing data drove the uncertainty of model simulated estimates (i.e., precipitation, evaporation, and runoff) over High Mountain Asia due to significant differences in the quality of reliable reference measurements over the domain."

Specific comments – It is a bit surprising to me finding in the areas with deeper snowpack the largest uncertainties, since simulating shallow snowpacks is often more challenging than deeper ones.

We agree that simulating shallow snowpacks is often more challenging due to their sensitivity to changes in air temperature, influence of substrate conditions and wind redistribution processes that are typically not included in LSMs. From our results, there is a greater spread in absolute SWE in the deeper snowpacks (in the mountains) where the average SWE and standard deviation between ensemble members are largest. However, we see large uncertainty in terms of normalized SWE across a large portion of the domain (Figure 3b), and there is a greater spread in the total water storage (SWS) in the shallow snow areas that accounts for differences in the aggregated snow estimations over the large areal extent.

I would like to hear the hypothesis of the authors about this result. Are the climate (i.e. precipitation) in these areas more difficult to be simulated?

Yes. The coarser resolution atmospheric models generally do not simulate enough snowfall in the mountains due to their inability to resolve the steepness of the topography.

We've discussed this in lines 378-379 on page 12 as follows: "The coarser resolution atmospheric models generally do not simulate enough snowfall in the mountains due to their inability to resolve the complexity of the topography (Lundquist et al., 2019)."

Is it more/less affected to uncertainties in snow-rain separation?

A static environmental lapse rate of 6.5 K/km from NLDAS1 and NLDAS2 projects is used to apply an elevation adjustment to the coarse meteorological fields. We acknowl-

edge this is not perfect and doesn't work everywhere, but it is a standard approach used in other products. This constant factor could be a source of uncertainty in the rain-snow transition, particularly over mountainous areas. The partitioning schemes used by the models (simple temperature threshold versus transition temperature range) also play a role.

These issues are described in lines 248-253 on page 8 as follows: "These highly complex terrains have relatively high snowfall precipitation, and the large spread is partially due to different rain/snow partitioning schemes in each LSM. While Noah2.7.1, JULES, and CLSMF-2.5 use a simple temperature threshold of 0oC to distinguish rainfall and snowfall precipitation, Noah-MP3.6 includes a transition temperature range described in Jordan (1991) (see Table S1). While our lapse-rate correction method is based on approaches used in other products (see Section 2.3), the lack of considerations of spatial variability in the snow-rain partition is a limitation, particularly over mountainous areas."

Same for different uncertainty between flat areas and mountains (Forcing datasets have less observations in mountains?

According to NOAA National Weather Service's National Operational Hydrologic Remote Sensing Center (NOHRSC) National Snow Analyses, 61% of snow observations (e.g., SWE and snow depth) are located in mountainous areas (defined as a landform that rises at least 1,000 feet) in the U.S. On the other hand, 91% of met stations (e.g., snowfall stations) are located in flat areas. Over Canada, there are fewer available ground and radar measurements than in the continental U.S., which could lead to larger differences in terms of input meteorology.

The uncertainty in mountains is associated to the downscaling technique, or again the reproducibility of the climate there?

Both have implications and these are areas we are hoping to focus future efforts on.

We have briefly discussed this in lines 361-363 on page 12 as follows: "Previous studies also highlighted the limitations of coarse-resolution models, particularly in capturing snow accumulation in mountain areas, and suggested using a resolution of <10 km (Ikeda et al., 2010; Kapnick & Delworth, 2013; Pavelsky et al., 2011; Wrzesien et al., 2017)."

and in lines 378-379 on page 12 as follows: "The coarser resolution atmospheric models generally do not simulate enough snowfall in the mountains due to their inability to resolve the complexity of the topography (Lundquist et al., 2019)."

If the individual members are compared to the reference snow datasets, the complexity of the snow-vegetation parametrization could be used to explain results when forested and non-forested areas are compared?

Thanks for this suggestion. Different snow-vegetation parameterization was used in each LSM (see Table S1 in supporting information). For example, Noah-MP3.6 uses a dynamic vegetation model and semi-tile vegetation sub-grid scheme, but Noah2.7.1 doesn't have both. We are planning to make an evaluation of this parameterization a focus of future efforts, but choose to exclude the details of that discussion in order to maintain a reasonable manuscript size.

The use of 1 or several layers for snowpack could be also identified as potential source of uncertainty shown in the study.

We agree with your comment. If we compare individual members, the snow layering scheme could be a source of uncertainty. Again, we are trying to differentiate this work from previous SnowMIP studies and considering the assessment of uncertainty of the individual members of the ensemble as a new and novel manuscript advancement.

- Line 73: 5 km instead of 5km

We have corrected this.

- Line 207: "Section 3.1 compares the ensemble with observations derived from data

assimilation techniques" I would not call them observations, perhaps is better using comparison with the reference snow datasets

We have corrected this sentence (in lines 214-215 on page 7).

Section 3.2.2. Not sure if seasonal variability is the most accurate tittle for the subsection, what about: "Timing of annual peak SWE. . ."

We have revised the title to "Timing of annual peak SWE"

- Figure 5 and 6: Is it possible to add boxplots with the values for the snow datasets used as references? It can give a good indication about some specific LSM or forcing dataset clearly biased from "reality".

Thanks for suggesting this. However, it's difficult to add those for the snow datasets due to their limited spatial coverage. For example, UA and SNODAS only cover the CONUS. CMC only has snow depth information.

-It would be also good to show a figure as 5 (probably as supplementary) but specifically for areas where the annual behavior of snow patterns is known to be well contrasted (often with opposite anomalies; i.e. Cascades and Rockies US) because the average over the entire domain smooths the differences and lead to very little interannual variability.

Thanks for this suggestion. We've added Figure S1 in supporting information and the following text in lines 283-287 on page 9.

"Figure S1 shows two time series of domain-averaged daily mean SWE of the Rockies Mountains and the Cascades in the United States (See Fig. 1b and Section 3.3.1) where the annual snow behavior is known to be well contrasted (Marshall et al., 2019). In the U.S. Rockies, the spread across the ensemble is smaller and the annual maximum SWE is relatively unchanged as compared to those of higher elevations in the Cascades."

- The subsection Section 3.2.5 directly links higher uncertainty with lack of observations. I am not sure if this statement is sound because the reasons of the uncertainties are not well identified along the manuscript.

We agree that the causes of uncertainty warrant greater inspection, and we hope to focus future efforts on these questions. We still feel that this analysis provides insight as to when and where spaceborne observations may help improve SWE and SWS estimation, regardless of the cause. We have added the following sentence in lines 306-308 on page 10 to acknowledge the limitations of the existing study:

"While additional analysis is needed to understand and improve the model parameterizations that are driving the ensemble spread, remote sensing observations have the potential to reduce uncertainty in global SWE and SWS estimation."

Further, lines 506-509 on page 16 adds: "We acknowledge that the influence of the sources of uncertainty such as the choice of model parameters and spatial resolution of topographic features are not examined in this study. Additional work is needed to understand the specific drivers of uncertainty within model physics, better characterize the snow storage over mountain and non-mountainous regions, and quantify the regional influence of SWE uncertainty on water availability."

-Line 368: referring to GRACE: "total terrestrial water storage (TWS) anomaly observations showed reasonable results (not shown)." I do not understand what authors really mean.

We agree it's confusing and have removed this from the text.

- Caption of Figure 8 says "Rockies Canaidan"

The caption has been corrected.

- Section 3.4 is interesting and indeed the topic could be a new manuscript itself. However, as it is presented I have the feeling that many of the conclusions derived from this section are not fully supported needing deeper analyses. Authors may consider

remove this section that could compensate to develop more other sections (i.e., assessment of uncertainty of the individual members of the ensemble).

Thanks for this suggestion. We agree that this section is describing a preliminary analysis that requires further exploration. We think this section helps to set up the discussion on required future work and the need for improved snow data to improve streamflow estimation. We've tried to explain explicitly this in lines 461-464 on page 15.

"While this is a preliminary analysis that requires further exploration, it helps to provide insight into the need for improved snow data to improve streamflow estimation. A more detailed examination of the influence of SWE-runoff uncertainties, an investigation into the utility of SWE observations to reduce SWE uncertainty, and thereby runoff uncertainty, are left for future work."

References

Essery, R., Rutter, N., Pomeroy, J., Baxter, R., Stahli, M., Gustafsson, D., Barr, A., Bartlett, P. and Elder, K.: SnowMIP2: An evalution of forest snow process simulation, Bulletin of the American Meteorological Society, doi:10.1175/2009BAMS2629.1, 2009.

Ikeda, K., Rasmussen, R., Liu, C., Gochis, D., Yates, D., Chen, F., Tewari, M., Barlage, M., Dudhia, J., Miller, K., Arsenault, K., Grubišić, V., Thompson, G. and Guttman, E.: Simulation of seasonal snowfall over Colorado, Atmospheric Research, doi:10.1016/j.atmosres.2010.04.010, 2010.

Kapnick, S. B. and Delworth, T. L.: Controls of global snow under a changed climate, Journal of Climate, doi:10.1175/JCLI-D-12-00528.1, 2013.

Lundquist, J., Hughes, M., Gutmann, E. and Kapnick, S.: Our skill in modeling mountain rain and snow is bypassing the skill of our observational networks, Bulletin of the American Meteorological Society, doi:10.1175/BAMS-D-19-0001.1, 2019.

Jordan, R.: A one-dimensional temperature model for a snow cover: Technical documentation for SNTHERM.89., 1991.

Marshall, A. M., Abatzoglou, J. T., Link, T. E. and Tennant, C. J.: Projected Changes in Interannual Variability of Peak Snowpack Amount and Timing in the Western United States, Geophysical Research Letters, doi:10.1029/2019GL083770, 2019.

Pavelsky, T. M., Kapnick, S. and Hall, A.: Accumulation and melt dynamics of snowpack from a multiresolution regional climate model in the central Sierra Nevada, California, Journal of Geophysical Research Atmospheres, doi:10.1029/2010JD015479, 2011.

Rutter, N., Essery, R., Pomeroy, J., Altimir, N., Andreadis, K., Baker, I., Barr, A., Bartlett, P., Boone, A., Deng, H., Douville, H., Dutra, E., Elder, K., Ellis, C., Feng, X., Gelfan, A., Goodbody, A., Gusev, Y., Gustafsson, D., HellströM, R., Hirabayashi, Y., Hirota, T., Jonas, T., Koren, V., Kuragina, A., Lettenmaier, D., Li, W.-P., Luce, C., Martin, E., Nasonova, O., Pumpanen, J., Pyles, R. D., Samuelsson, P., Sandells, M., SchäDler, G., Shmakin, A., Smirnova, T. G., StäHli, M., StöCkli, R., Strasser, U., Su, H., Suzuki, K., Takata, K., Tanaka, K., Thompson, E., Vesala, T., Viterbo, P., Wiltshire, A., Xia, K., Xue, Y. and Yamazaki, T.: Evaluation of forest snow processes models (SnowMIP2), Journal of Geophysical Research: Atmospheres, 114(D), D06111, doi:10.1029/2008JD011063, 2009.

Wrzesien, M. L., Durand, M. T., Pavelsky, T. M., Howat, I. M., Margulis, S. A. and Huning, L. S.: Comparison of methods to estimate snow water equivalent at the mountain range scale: A case study of the California Sierra Nevada, Journal of Hydrometeorology, doi:10.1175/JHM-D-16-0246.1, 2017.

Yoon, Y., Kumar, S. V., Forman, B. A., Zaitchik, B. F., Kwon, Y., Qian, Y., Rupper, S., Maggioni, V., Houser, P., Kirschbaum, D., Richey, A., Arendt, A., Mocko, D., Jacob, J., Bhanja, S. and Mukherjee, A.: Evaluating the uncertainty of terrestrial water budget components over high mountain asia, Frontiers in Earth Science, doi:10.3389/feart.2019.00120, 2019.

[Figure]

[Figure]

[Figure]

Figure S1: Time series of domain-averaged mean SWE of Rockies, USA and Cascades. Different colors and line style were used to represent each ensemble; a bold black solid line represents the domain-averaged ensemble mean; the units are mm.

**Fig. 1.**

---

## Author Comment (AC3) · 10 Dec 2020

Dear Referee,

Thank you for your valuable comments. Below we have provided a response to each of the comments provided by the referee.

Anonymous Referee 3's comments: The authors present a highly interesting and relevant study assessing snow modelling uncertainties across the North American region with (among others) the goal of providing information about global snow observation

needs. The study comprises four different land surface models and three forcing data sets, resulting in an ensemble of twelve members. The results show that the uncertainty represented by the ensemble spread varies across the study domain, with for instance high uncertainty in the simulation results for mountainous and forested regions. The authors conclude that for these regions high-resolution observations are needed to capture the high spatial variability in snow water equivalent. Overall, the article is very well written, easy to follow and from the technical perspective ready for publication in my opinion. Nonetheless, the current manuscript only provide rather shallow information about what snow observations are required in order to reduce the uncertainty seen in the simulation results. What variables should be measured and on which spatial and temporal resolution? For example, the need for high-resolution observations in mountains to capture the high spatial snow water equivalent variability has been recognized a long time ago. However, for very large domains, we still lack observations with sufficient quality as well as high enough spatial and temporal resolution, and this may not change soon either. Providing some more details, foremost more quantitative, about the observational needs to constrain the model uncertainties would make the paper even more interesting to read.

We agree with these excellent points that more detailed requirements of snow observations would be a nice outcome of this study. In fact, the SEUP efforts are designed with this quantification in mind, and this paper represents the initial phase that establishes a baseline of the snow modeling uncertainty over the continental scales. In this paper, we focus on providing guidance on when and where the snow modeling uncertainties are highest. Following-up on this effort, our group has been working on SEUP Phase II, to conduct observation system simulation experiments (OSSEs) that systematically quantifies the worth of the existing data and data yet be collected, and recently, these techniques have been applied to snow modeling studies. The snow OSSE derives the utility from a multitude of observations by exploring the interaction between different observation types and observation sampling methods (e.g., using different spatial and temporal resolutions). The results from this effort will be evaluated across a suite

of metrics that capture the information of the observations and uncertainty and are expected to help in the choice and refinement of sensors for the accurate characterization of global, terrestrial snow mass.

We have added the following sentence in lines 509-513 on pages 16-17 to acknowledge the focus of future efforts:

"More detailed requirements of snow observations (e.g., choice of observation types and sampling methods) will be focused in future efforts by conducting observation system simulation experiments (OSSEs) that systematically quantify the worth of the existing data and data yet to be collected. The results from this future effort are expected to help in the choice and refinement of sensors for the accurate characterization of global, terrestrial snow mass."

---

## Author Response (AR2)

**Dear Dr. Essery**

*Thank you for your comments. Based on the feedback, we have made several changes and improvements to the manuscript. Below we have provided a response to each of the comments provided. The reviewer's original comments are in black and our responses are in red.*

178
State that UA is at 4 km spatial resolution

We have revised this sentence in lines 177-178 on page 6 as:
"(2) daily gridded estimates of snow depth and SWE developed by University of Arizona (UA; Zeng et al., 2018) available at 4 km spatial resolution,"

204
The Sturm et al. (1995) classification includes an Alpine class that does not appear in Figure 1c. The updated Liston and Sturm (2014) classification is unpublished and inaccessible.

We agree it's confusing and have removed 'Strum et al. (1995)' and revised this sentence as follows:
"In this analysis we use a snow classification at a higher (10 km) resolution proposed by Liston and Sturm (2014, unpublished), which analyzes the relationships among textural and stratigraphic characteristics of snow layers, climate variables (e.g., air temperature, precipitation, and wind speed), and vegetation to globally categorize terrestrial snow into seven classes: Tundra, Taiga, Maritime, Ephemeral, Prairie, Warm forest, and Ice."

210
MODIS-IGBP has 17 land cover classes

Thanks for correction. It's been corrected.

240
Delete ", as we will show in Section 3.2.2" (information repeated in the next sentence).

It was deleted.

291
"across the forcing datasets when driving a common LSM"

Corrected.

380
"change in total water storage from GRACE data"

Corrected.